# Disconnecting The Dots: Creating Leakage-Free Protein Datasets by Sparse Removal of Densely Connected Data Points

## Abstract

Biological systems arise through evolutionary processes that effectively render all biological data, at scales ranging from biomolecules to organisms, to be evolutionarily related. This poses a challenge to assessments of model generalization, as naive random splits do not safeguard against data leakage; all data points are in some sense related, and their degree of relatedness lies on a continuum. To address this challenge, various similarity metrics are typically used to cluster data prior to splitting to ensure dissimilarity of resulting partitions. However, as we show in this study, similarity thresholds that lead to well-behaved splits (large numbers of homogeneously sized clusters) must invariably be too permissive, thus only permitting assessment of weak generalization. Conversely, stringent thresholds that could in principle enable assessment of strong generalization typically fail to produce well-separated clusters, yielding one or a handful of very large clusters that span the entire dataset. Here, we propose a new data splitting methodology that optimally balances these competing considerations by relaxing the assumption that all data points must be retained. Instead, through a principled and judicious removal of highly central data points, our approach yields well-behaved data splits that enable assessment of extreme generalization regimes. We demonstrate its utility by investigating the impact of diverse proteins representations on protein function prediction. Our experiments confirm the robustness of our new methodology and provide insights into the utility and behavior of protein representations under previously untested regimes of sequence and structure generalization.

## 1 Introduction

Generalization to unseen data is a highly sought desideratum for machine learning models. It is often assessed by holding out data from the training set that can be used specifically for *post hoc* validation (Goodfellow et al., 2016). For data modalities conventionally used in machine learning, such as human text or natural images, random data splits can be sufficient. Biological data, however, has the property of being generated by a universal evolutionary process that connects all known life on Earth. Different genes, cell types, and organisms lie on a continuum of similarities with one another. When partitioned randomly, biological data points can become arbitrarily close across the training/validation/test set divide, thereby inflating reported generalization performance and, more generally, making it an uncontrolled downstream variable of the randomness of the underlying split.

To address this issue, clustering of biological data prior to splitting has become standard practice in bioinformatics. This raises the question of which similarity metric, and numerical threshold, should be used for clustering (*e.g.,* 30% sequence identity is a standard choice for proteins). These choices bring their own challenges however. If the similarity threshold is too permissive, the resulting splits can be too easy, only testing generalization to nearby regions of data space. On the other hand, if they are too strict, all data can collapse into one cluster, owing to the evolutionary relationships between data points. This makes it difficult to create balanced data splits that permit machine learning. This is particularly common when all forms of "leakiness" are sought to be avoided, often formally implemented using single linkage clustering. In such cases, the transitivity of evolution, where entity A is related to B and B to C, makes A and C fall into the same cluster. Common approaches seeking to prevent this by using more restrictive clustering approaches invariably lead to data leakage.

Here, we introduce a methodology for constructing data splits that meet three desiderata: leakage-free, well-balanced, and sufficiently separated to permit assessment of difficult generalization tasks. Our approach is also fast and efficient and thus applicable to large biological datasets. It comes at the cost of intentionally removing possibly high quality data points from all data partitions. Specifically, we judiciously remove central data points that are similar to many other data points, fragmenting the evolutionary structure of the underlying data space to generate well-separated clusters of homogeneous sizes. Partitioning data based on the resulting clusters results in well-behaved, leakage-free splits. This approach is agnostic to the similarity metric or threshold used, by trading data loss for improved partitions. We show under relevant testing regimes that the amount of data lost is minimal.

We demonstrate the utility of our approach by assessing the capacity of learned protein representations to capture sequence-structure-function relationships. We test generalization across both sequence and structure space and find the latter to be more challenging. Notably, the lowest sequence similarity threshold is as difficult as the highest structure similarity threshold. This confirms the hypothesis that traditional sequence splits can inadvertently leak data by missing evolutionary relationships that are only detectable structurally (proteins can have highly dissimilar sequences while adopting similar structures). Our study thus provides new insights into the behavior of protein representations. We also introduce a new approach for constructing biological datasets with applications ranging from pathogenicity prediction to drug design.

## 2 RELATED WORK

### 2.1 PREVENTING LEAKAGE BETWEEN DATA SPLITS

Leakage in biological data is a widely recognized problem (Whalen et al., 2022). Random splits have been shown to significantly inflate model performance, as demonstrated at both the single-sequence level (Rao et al., 2019; Dallago et al., 2021) and in protein-protein interactions (Park & Marcotte, 2012; Hamp & Rost, 2015; Bernett et al., 2024). However, the extent to which solving it becomes increasingly difficult or even impossible as higher generalization capabilities are sought is less well recognized. Multiple approaches to leakage prevention currently exist. One is temporal, training models on date released prior to a cutoff data and evaluating on data released after it. CASP (Critical Assessment of protein Structure Prediction) keeps target structures unreleased until after model submissions (Kryshtafovych et al., 2023). This approach is also common among protein-ligand modeling methods such as EquiBind (Stärk et al., 2022), TankBind (Lu et al., 2022), DiffDock (Corso et al., 2022) and PoseBusters (Buttenschoen et al., 2024). While easy to implement, it does not account for evolutionary similarity among proteins or chemical similarity among ligands.

Another approach uses known biophysical properties as a proxy for evolutionary or chemical similarity. For instance, PoseBusters filters proteins and ligands based on their molecular weight or atom content. DeepChem (Ramsundar et al., 2019) proposes data splits for small molecules based on their fingerprints, scaffolds, and weights. Others directly rely on existing annotations of evolutionary relationships between data points, facilitating splits based on known clusters. This approach is common among protein modeling methods (Ingraham et al., 2019; Anand et al., 2022). Widely used classification schemes include ECOD (Evolutionary Classification of protein Domains) (Cheng et al., 2014) and CATH (Orengo et al., 1997), which derive evolutionary relationships based on sequence and structure, respectively. The approach can be effective at preventing leakage but often incurs substantial data loss, as it is constrained by the size of the annotation database.

The perhaps most common approach is to explicitly perform an initial clustering step with a similarity metric reflective of the data modality in question. Widely used protein clustering algorithms include CD-Hit (Fu et al., 2012) and MMseqs2 (Steinegger & Söding, 2017) for sequences and Foldseek (van Kempen et al., 2023) for structures. A representative exemplar of this approach is MaSIF-site (Gainza et al., 2020; Sverrisson et al., 2021; Gainza et al., 2023), which clusters protein sequences at 30% sequence identity and retains only the cluster representatives for training and testing. While the approach effectively prevents leakage, it also discards 75% of the data. ProteinNet (AlQuraishi, 2019) carries out the same type of clustering at varying levels of sequence identity but is only applicable to data in the Protein Data Bank (Berman et al., 2000a). In contrast, Deep-FRI (Gligorijević et al., 2021) distributes entire protein clusters between the training and test sets, while FLIP (Dallago et al., 2021) uses both complete clusters and cluster representatives to construct benchmarking datasets. DataSAIL (Joeres et al., 2023) formulates data splitting as a set of

binary linear optimization problems to minimize similarity between splits, and applies its solution to both single sequences and sequence clusters. Although relatively slow due its complexity and lacking guarantees for strict separation, DataSAIL has the advantage of handling both 1D (e.g single molecules) and 2D (e.g. protein-ligand pairs) data.

## 2.2 Breaking evolutionary relationships

The fundamental problem unsolved by these prior approaches is the dense interconnectivity of the underlying data, which becomes increasingly apparent with larger datasets and more stringent similarity thresholds. For example, in ProteinFlow (Kozlova et al., 2023), proteins are clustered by sequence identity, forming nodes in a graph connected by edges if a protein complex spans proteins from different nodes. The connected components of the graph define the final set of clusters. However, one large component, comprising approximately 20% of the dataset, is left unpartitioned, making it difficult to create representative train/test splits. CCPart (Fernández-Díaz et al., 2024) also finds that connected component clustering typically produces one dominant cluster with most sequences, but leaves it intact. Smaller clusters are assigned to the evaluation set, skewing it towards the most disconnected regions of the protein space. While these approaches ensure strict separation between clusters, essential for avoiding data leakage, they fail to address the imbalance caused by the large, unpartitioned component.

To achieve well-balanced and disconnected splits, two recent approaches have introduced splitting algorithms that remove sequences from the dataset. After an initial clustering step based on sequence identities calculated with MMseqs2 (Steinegger & Söding, 2017) or EMBOSS (Rice et al., 2000), GraphPart (Teufel et al., 2023) iteratively assigns clusters to partitions. It then disconnects them by iteratively reassigning or removing similar sequences across partitions according a heuristic. Although this method enforces strict homology separation, it often leads to substantial or even complete data loss, particularly at low sequence identity thresholds.

In the small molecule context, Lo-Hi (Steshin, 2023) shares our objective of breaking large clusters of related entities. Specifically, it offers a solution based on Integer Linear Programming to the Balanced Vertex Minimum k-Cut problem, an NP-hard problem (Cornaz et al., 2014; Balas & Souza, 2005; Schwartz, 2022) which aims to disconnect a graph into k partitions (e.g train and test) of predefined sizes by removing the fewest possible nodes. While conceptually well-suited to the challenge of disconnecting the sequence space into disjoint splits, the method tackles a complex optimization problem and does not always converge to a solution. Moreover, the graph coarsening step employed to speed up computations may cause information loss.

Like these approaches, our method starts with calculating similarity values for all pairs in the dataset. Unlike them, however, we do not seek optimal cuts in the protein graph. Instead, we apply community detection to identify biologically meaningful, densely connected groups of proteins. We then iteratively disconnect these communities by removing the node in the largest cluster with the most inter-community connections until all inter-community connections are gone, following the "remove until done" strategy from the second homology reduction algorithm in Hobohm et al. (1992). Central to our approach is the idea that clusters preserve the structure of the underlying data. While likely not theoretically optimal, the simplicity of our method, coupled with its implementation using sparse numpy matrices, make it both extremely fast and effective in practice, with minimal data loss.

Additionally, similar to ProteinNet but unlike GraphPart and Lo-Hi, our method allows to include clusters at varying similarity thresholds in the validation and test sets, eliminating the need to re-run the pipeline for each threshold and without significantly increasing compute time. This enables direct comparisons across different levels of generalization difficulty.

## 2.3 Sequence vs. structure-based metrics in protein data

Agnostic to the splitting strategy is the choice of similarity metric used to compare data points. In this work we apply our methodology to the problem of protein function prediction, which requires splitting data based on protein similarity metrics. To date, most assessments of protein models are done based on sequence-based splits, even for celebrated protein structure prediction methods (Abramson et al., 2024). This trend stems from the abundance of protein sequences in databases (uni, 2023) and sensitive profile-based search methods such as JackHMMer (Eddy, 2011) and MMseqs,

which make it possible to detect very low levels of sequence similarity (>25% sequence identity). More recently, SpanSeq (Ferrer Florensa et al., 2024) introduced a method for calculating a proxy for sequence similarity for very long sequences through fast k-mer comparisons. In contrast, the historical paucity of protein structures combined with the computationally demanding nature of structure-based alignment methods such as TMAlign (Zhang & Skolnick, 2005) have limited the adoption of structure similarity metrics.

Structures, however, are more conserved than sequences (Illergård et al., 2009), such that proteins with very dissimilar sequences (substantially below 25% sequence identity) can be structurally similar. Generalization across structure can thus be more challenging than generalization across sequence, and is especially pertinent for protein function prediction given the role that structure plays in function. This also makes it an excellent testbed for our methodology. The dense connectivity of protein space becomes most apparent when using structure-based metrics, making it difficult-to-impossible to use with existing data partitioning methodologies if data leakage is to be avoided.

A recent study identified such leakage in protein interaction benchmarks (Bushuiev et al., 2024). Using sequence-based partitioning of protein complexes similar to ProteinFlow (Kozlova et al., 2023), the study found proteins with analogous interfaces and folds distributed across splits. The authors recommended adopting structure-based splitting instead. This was done by ProteinShake (Kucera et al., 2024) using structure-based clustering performed by Foldseek (van Kempen et al., 2023). Other recent approaches include MaSIF-search (Gainza et al., 2020), which generated non-redundant protein surface interaction datasets, and iDist (Bushuiev et al., 2023), which identified near-duplicate protein-protein interfaces through graph representations. In these studies, structural generalization is assessed at single and somewhat low difficulty levels. PINDER (Kovtun et al., 2024) combined both sequence and structure similarity metrics using MMseqs and Foldseek to minimize interface similarity between splits. Similarly, PLINDER (Durairaj et al., 2024) proposed protein-ligand interaction splits by integrating similarity metrics across multiple levels (protein, pocket, ligand, and interaction) for both protein and ligand sequences and structures.

# 3 DATASET CONSTRUCTION METHOD

## 3.1 ASSUMPTIONS AND SETUP

We begin by assuming that an appropriate similarity metric exists for our dataset and task of interest. We emphasize that our approach is agnostic to the specific choices made, although in practice the final results will be sensitive to them. We construct a graph $G$ to encode the data, where nodes correspond to data points and edges to similarity scores of the nodes they connect. Because of the dense connectivity of the underlying data space, even a binarized version of the graph, where edges are removed below a certain threshold, would still result in a very large connected component. Thus a naive but leakage-preventing clustering of the dataset would not yield usable splits. Our task then is to identify a small number of nodes to remove such that the graph is disconnected into a sufficiently larger number of homogeneously-sized components to permit the construction of training/validation/test splits.

## 3.2 REMOVAL OF CENTRAL POINTS

Graph partitioning algorithms such as Kernighan–Lin (Kernighan & Lin, 1970) can find balanced partitions in a graph under the constraint of minimizing the number of edges cut between nodes. However, they do not provide a straightforward way to choose a minimal set of nodes to remove to disconnect the graph into separate components. Finding this optimal set of nodes to remove to obtain well-balanced, strictly separated partitions is NP-hard (Cornaz et al., 2014; Balas & Souza, 2005; Schwartz, 2022; Cornaz et al., 2019). There is also no guarantee that the resulting graph components are densely connected.

Instead of successively partitioning $G$ into smaller components, we adopt a bottom-up approach (Algorithm 1) which we illustrate in Figure 1A. We start by employing the Leiden algorithm (Blondel et al., 2008; Traag et al., 2019), a method commonly used for analyzing large networks, to find communities $C$ within $G$. The number and size of communities can be tuned using the resolution parameter $r$. These communities are not disconnected from one another but are optimized to have higher connectivity within their nodes than with external nodes. Next, we iteratively select the

largest community $c$ and remove the node $v$ in $c$ that has the most inter-community edges. When there are no more inter-community edges in $G$, we finally extract $G$'s connected components $K$. Systematically removing top connectors allows us to sever as many inter-community connections as possible while removing only one node at a time. Additionally, focusing on the largest community to remove highly central nodes helps us break the largest clusters first, while preserving the structure of smaller communities. These two considerations enable us to efficiently disconnect the communities with minimal data loss.

Our algorithm also has the advantage of being fast. The Leiden algorithm employs a greedy optimization method, as does the process of iteratively removing nodes with high betweenness centrality, which can become a bottleneck for large-scale datasets. To accelerate runtime, our implementation exclusively perform computations over sparse matrices. In our experiments, encoding the adjacency matrix $M$ as a sparse array reduced the runtime from over an hour to just a few minutes for a dataset of approximately 30,000 data points.

---

**Algorithm 1** Identify and disconnect communities

---

1: **Inputs:** $M$: adjacency matrix of similarity scores, $r$: resolution for community detection
2: **Outputs:** $K$: connected components, $G$: graph, $M$: adjacency matrix
3:
4: **def** DisconnectCommunities($M$: Array, $r$: float, $thr$: float) $\rightarrow$ (List[int], Graph, Array):
5:     Construct graph $G$ from adjacency $M$ at threshold $thr$
6:     # Identify communities $C$
7:     $C \leftarrow$ Leiden algorithm($r$)
8:     # Disconnect communities $C$
9:     **while** inter-community edges **do**
10:       Find largest community $c$ in $C$
11:       Find node $v$ with most inter-community edges in $c$
12:       Remove $v$ from $G$ and $M$
13:     **end while**
14:     $K \leftarrow$ connected components from $G$
15:     **return** $(K, G, M)$

---

### 3.3 GENERATING DATA SPLITS

After clustering the data, we create data splits by sampling clusters for the validation and test sets at increasing similarity thresholds $T$. We describe sampling procedure in Algorithm 2 and illustrate it in Figure 1B. We essentially follow ProteinNet's methodology. We cluster the data at the lowest similarity threshold $T[0]$ and randomly sample a fixed number $n$ of clusters, which we split equally between the validation and test sets. We then repeat the process at the next higher threshold. Finally, the remaining proteins at the highest threshold constitute the training set. The key conceptual difference with ProteinNet is that we replace the initial clustering step with our own community-based clustering approach. We find that performing the community detection and disconnection step once at the lowest similarity threshold is sufficient to break the connectivity of the underlying evolutionary space, eliminating the need to repeat it at higher thresholds.

## 4 EXPERIMENTAL SETUP

### 4.1 FUNCTION PREDICTION TASK

We apply our community-based clustering approach to the two datasets introduced by DeepFRI (Gligorijević et al., 2021), both derived from the Protein Data Bank (Berman et al., 2000b) and commonly used to benchmark protein function prediction models (Lai & Xu, 2022; Zhang et al., 2022; 2023). The **Gene Ontology (GO) dataset** (36,640 proteins) includes functional annotations across three distinct ontologies: Biological Process (BP: 1,943 annotations), Cellular Component (CC: 320), and Molecular Function (MF: 489). Models for each ontology are trained separately. The **Enzyme Commission (EC) dataset** (19,198 proteins) focuses on predicting catalytic activity

---

**Algorithm 2** Generating data splits

---

1: **Inputs:** $M$: adjacency matrix of similarity scores, $T$: similarity thresholds, $r$: resolution for community detection, $n$: number of clusters sampled per threshold for validation and test
2: **Outputs:** $D$: leakage-free data splits at thresholds $T$
3:
4: **def** SampleDataSplit($M$: Array, $T$: List[float], $r$: float, $n$: int) → Dict:
5:     $D \leftarrow \{\}$
6:     # Sort thresholds from lowest to highest
7:     $T \leftarrow \text{sorted}(T)$
8:     # Do community detection at the lowest threshold
9:     $thr \leftarrow T[0]$
10:     Remove edges $\leq thr$ in $M$
11:     $(K, G, M) \leftarrow \text{DisconnectCommunities}(M, thr, r)$
12:     $val_{thr}, test_{thr} \leftarrow$ randomly sample $n$ clusters from $K$
13:     $D[thr] \leftarrow (val_{thr}, test_{thr})$
14:     Remove sampled nodes from $G$ and $M$
15:     # Iteratively sample clusters at increasing thresholds
16:     **for** $thr$ **in** $T[1:]$ **do**
17:         Remove edges $\leq thr$ in $G$ and $M$
18:         $K \leftarrow$ connected components from $G$
19:         $val_{thr}, test_{thr} \leftarrow$ randomly sample $n$ clusters from $K$
20:         $D[thr] \leftarrow (val_{thr}, test_{thr})$
21:         Remove sampled nodes from $G$ and $M$
22:     **end for**
23:     $D[train] \leftarrow$ remaining nodes
24:     **return** $D$

---

of enzymes at the third and fourth levels of the EC classification, with each protein assigned 538 labels. All tasks are framed as multiple binary, non-exclusive classification tasks.

## 4.2 SEQUENCE & STRUCTURE-BASED SPLITS

The DeepFRI datasets were originally split into approximately 80% for the training set, and 10% each for the validation and test sets. These splits were created by clustering protein sequences at various sequence identity thresholds (30%, 40%, 50%, 70% and 95%) using CD-Hit. We introduce novel splits based on sequence identity and the TM score, a measure of global structural similarity (Zhang & Skolnick, 2004; Xu & Zhang, 2010), to compare generalization across sequence and structure space. Pairwise sequence identities are calculated using MMseqs, where sequence identity is defined as the number of identical aligned residues divided by the number of aligned columns, including internal gaps (`--alignment_mode 3`). Structural alignments and TM scores are obtained using Foldseek's implementation of TM-align (Zhang & Skolnick, 2005) (`--alignment_type 1`).

For each similarity metric, we randomly assign clusters to the validation and test sets at five thresholds, between 30%-90% for sequence identity and 50%-90% for the TM score. We exclude lower thresholds because 30% sequence identity and 50% TM roughly correspond to the level where proteins are assumed to become randomly similar, and below which MMseqs and Foldseek lose sensitivity. We construct 20 splits per dataset (EC and GO) and similarity metric, evaluating models on the top 10 splits with the closest 80:10:10 train-validation-test ratio. For any dataset and similarity metric, results are averaged over these 10 splits. More details on split construction and statistics are provided in Appendix I.1.

## 4.3 METRICS FOR PROTEIN FUNCTION PREDICTION MODELS

Model performance is evaluated using two standard metrics from CAFA (Critical Assessment of Functional Annotation) (Radivojac et al., 2013; Jiang et al., 2016; Zhou et al., 2019): the protein-centric $F_{max}$, and the label-centric area under the precision-recall curve (AUPRC). The $F_{max}$ score, defined as the harmonic mean of precision and recall at an optimal prediction threshold, measures how well labels are assigned to proteins. AUPRC evaluates how well proteins are assigned to labels.

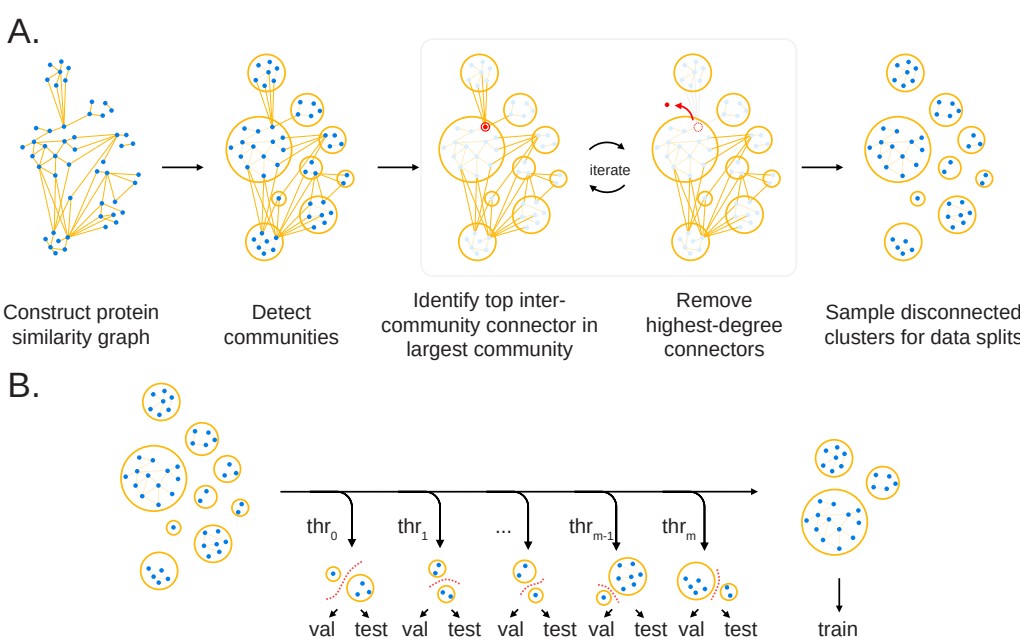

Figure 1: Overview of the dataset construction method. (A) We cluster data by representing the dataset as a graph, where nodes represent data points, and edges are weighted by their similarity score. We identify communities of highly related points and iteratively disconnect them by removing top connectors. (B) Strategy for sampling data splits for increasing similarity thresholds $thr_i$. At the lowest threshold $thr_0$, we cluster proteins using our community-based based approach. Next, we iteratively sample clusters for the validation and test sets by extracting the graph connected components. The remaining points at the highest threshold $thr_m$ constitute the training set.

Following (Gligorijević et al., 2021), we compute $F_{max}$ and AUPRC by averaging precision and recall over all proteins and over all functional annotations, respectively. Additionally, we introduce a cluster-based variant of the Fmax score, called Fmax-cluster, to evaluate performance over clusters rather than individual proteins. Precision and recall are averaged across clusters, addressing cluster size imbalances, which are not controlled for in our data splits. Formulas are provided in Appendix F.1.

## 4.4 MODELS

Our dataset construction pipeline provides an ideal framework for probing protein representations under different generalization regimes. We benchmark representations from two state-of-the art protein language models, ESM2-650M (Lin et al., 2022) and Ankh (Elnaggar et al., 2023), from structure prediction model AlphaFold (Jumper et al., 2021), from graph-based protein structure encoder GearNet-Edge (Zhang et al., 2022), and from sequence and structure-based encoder ESM-GearNet. Details on how these representations were obtained are provided in Appendix F.2. We find that training small, single-layer neural networks yields results comparable to or slightly better than previously reported state-of-the-art results on the original sequence-based DeepFRI splits (Zhang et al., 2023). Models are initialized with a random seed, and results are aggregated over both data splits and model seeds, with error bars representing the standard deviation. We train models until convergence and select the best checkpoint for evaluation by averaging the $F_{max}$ and AUPRC scores across all similarity thresholds. A distinct set of hyperparameters is tuned for each protein representation on the original DeepFRI split and applied to all new splits. Details on our hyperparameters are provided in Appendix F.3. Figure 2 summarizes our experimental setup.

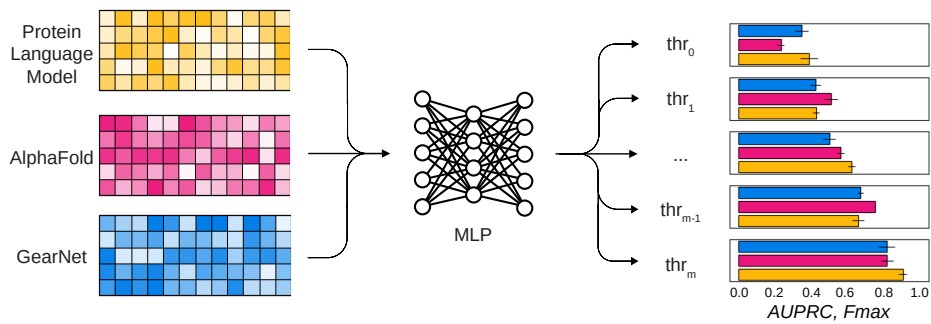

Figure 2: Overview of the experimental setup. We train simple feedforward neural networks on various sequence and structure-based protein representations, and evaluate them on novel sequence and structure-based across increasing levels of generalization difficulty, indicated by $thr_i$.

## 5 RESULTS ON THE DATASET CONSTRUCTION

### 5.1 DATA LEAKAGE FROM TRADITIONAL CLUSTERING

In Figure 5, we assess the proportion of "leaky" proteins in the original DeepFRI test sets, defined as those with a higher sequence identity to proteins in the training set than the maximum threshold. At low sequence identity levels, this proportion is remarkably high, with nearly 80% of leaky proteins in the EC dataset and 55% in the GO dataset at the 30% sequence identity threshold. Even at the highest threshold of 95%, a notable proportion of test proteins remain leaky. Figure 6 demonstrates that leaky proteins are, on average, significantly more similar to the training set than permitted. For instance, in the GO dataset, the average sequence identity of leaky proteins at the 50% sequence identity threshold is 76%, indicating that they are 50% more related to the training set than expected. This results from creating the splits using CD-Hit clustering, which does not ensure strict separation between clusters.

### 5.2 BREAKING PROTEIN INTERCONNECTIVITY

Figure 3 shows how our community-based clustering approach effectively partitions the protein structure space at low similarity levels (TM score = 0.5). After disconnecting the graph based on communities, the previously large connected component is fragmented into clusters of homogeneous sizes. In the EC dataset, the size of the largest component decreases from 73% to 1% of the entire dataset at the 0.5 TM threshold, and from 27% to 0.4% at the 30% sequence identity threshold. Similarly, in the GO dataset, the size of the largest component drops from 64% to 1% at the 0.5 TM threshold and from 27% to 0.4% at the 30% sequence identity threshold. Figures 7 and 8 illustrate the impact of the community-based clustering at various thresholds of sequence identity and TM scores. The dominant component is larger in the structural space than in the sequence space, likely due to the continuous nature of the local geometry of protein structures, as previously noted for structures from the Protein Data Bank (Skolnick et al., 2009). The resulting cluster distributions, visualized in Figures 9 and 10, still follow an inverse power law distributions but exhibit clusters of comparable sizes, facilitating the creation of well-balanced splits.

### 5.3 LIMITED DATA LOSS

Unless typical clustering methods, our approach trades-off data loss for better-behaved, strictly separated splits. For the EC dataset, we lose an average of 1.4% of the data using sequence identity as the similarity metric and 12.4% with the TM score. Similarly, in the GO dataset, 1.4% of the data is removed under sequence similarity, and 11.1% under structural similarity. Data loss is higher for structural similarity but remains acceptable, allowing for sufficiently large splits for training and evaluation. Given the greater structural connectivity among proteins, it is inevitable that more proteins need to be removed to effectively partition the structure space. Data loss is comparable across

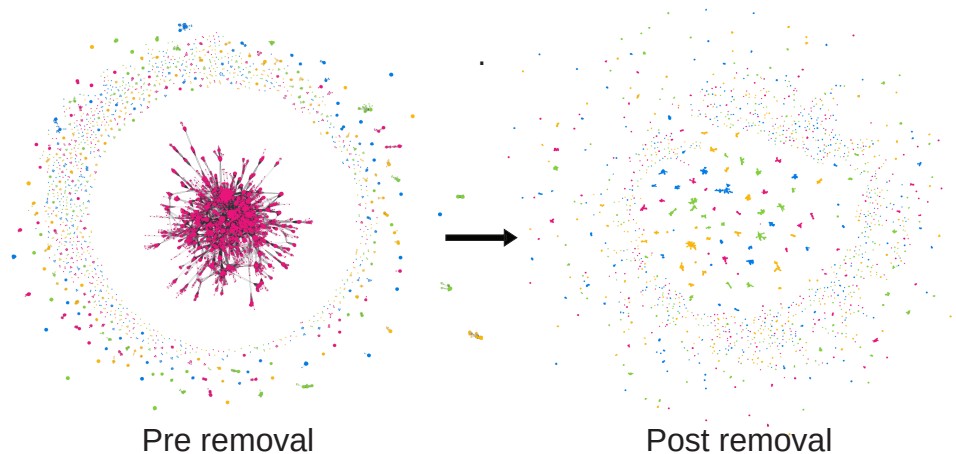

Pre removal          Post removal

Figure 3: Graph visualization of the protein space before (A) and after (B) partitioning it with our clustering method. The graph represents the enzyme (EC) dataset at the TM=50% threshold. Nodes correspond to proteins. Edges connect protein pairs when their TM score is above 50%.

datasets, likely due to both being derived from the same protein space, the Protein Data Bank. We also find that the sets of nodes removed by the community-based disconnection step is highly similar across random seeds, indicating that our method consistently removes the top connecting nodes, or hubs, in the data. Detailed statistics on the dataset splits can be found in Appendix I.2.

## 6   BIOLOGICAL INSIGHTS

### 6.1   GENERALIZATION IS HARDER IN STRUCTURE SPACE

The need for structure-based splits arises from the fact that proteins with highly similar structures can have vastly dissimilar sequences (Figure 12). Figure 13 shows a density plot illustrating the distributions of sequence identities and TM scores in the EC dataset. Notably, high-density regions appear both at high sequence identity and TM thresholds, as well as in areas where TM scores are high while sequence identities remain low.

Confirming our intuition that generalization to new structures is more challenging than to unseen sequences, Figures 4 and 14 illustrate model performance across various similarity thresholds. Our findings align with previous observations that structure-based splits are more difficult than sequence-based splits (Kucera et al., 2024). However, in our case, the difference in performance across similarity thresholds is more pronounced due to better separation of the evaluation sets. In contrast, model performance on the original DeepFRI splits (Figure 15) remains flatter across thresholds, indicating the presence of similar proteins within the sets at these thresholds.

Cellular Component shows a smaller drop in performance between sequence and structure splits, indicating that predicting cellular localization is largely independent of protein structure. For the other tasks, the highest TM threshold (0.9) nearly as challenging as the lowest sequence similarity threshold (30%), often regarded as the point below which proteins are randomly related. Performance at the 50%, 60% and 70% TM thresholds is very similar, indicating that models struggle to generalize to structures which are typically deemed similar - protein structures are usually considered randomly related below the 0.5 TM score (Zhang & Skolnick, 2005).

Additionally, we find that performance is consistently slightly lower when evaluated with $F_{max}$-cluster compared to Fmax (Figures 4 vs 14), suggesting that results may be overestimated when cluster effects are not accounted for. When calculated at the individual cluster level (Figures 16-18), Fmax scores significantly increase between the 0.7 and 0.8 TM thresholds, reflecting trends observed when results are averaged across clusters. Furthermore, the distribution of Fmax scores tends to be bimodal, indicating that some clusters are much harder than others, even at high thresholds. This disparity likely arises from the varying distances of these clusters are from the training set.

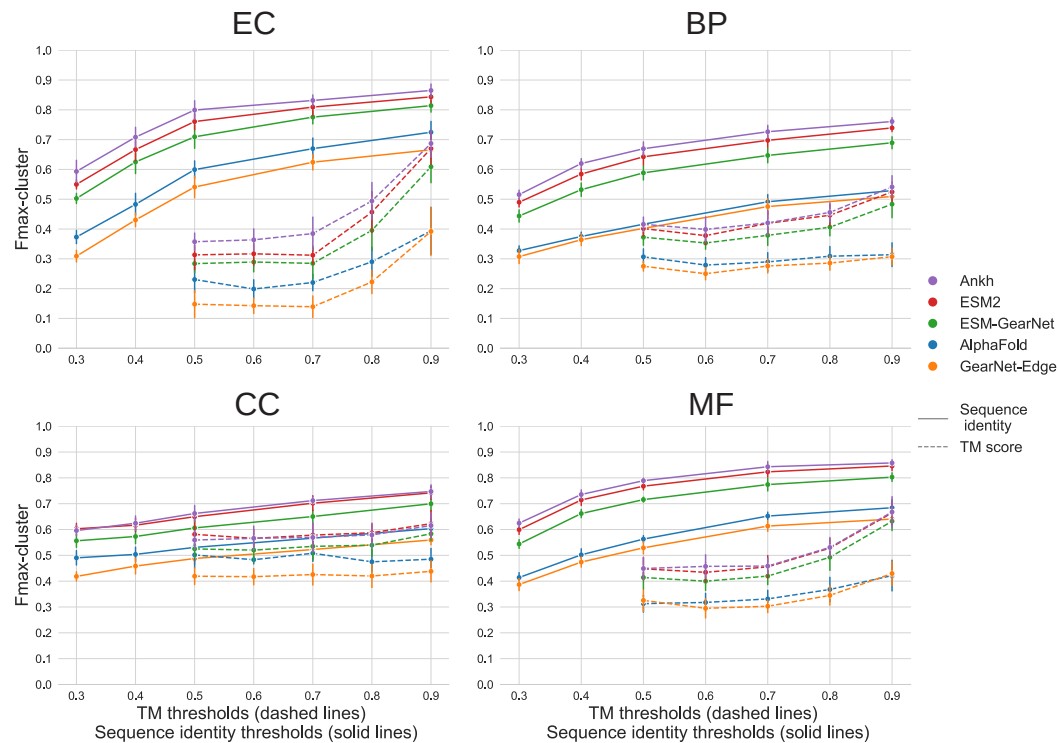

Figure 4: Results on the new sequence and structure splits with the $F_{max}$-cluster metric.

## 6.2 REPRESENTATIONS FROM LANGUAGE MODEL PERFORM CONSISTENTLY BETTER

As reported in Appendix G, our single-layer models trained on diverse protein representations achieve competitive results with state-of-the-art methods (Zhang et al., 2023). To the best of our knowledge, Ankh superior performance establishes a new state-of-the-art for the EC and MF ontologies. Interestingly, our results highlight the comparatively strong performance of protein language models across all similarity thresholds, whereas structure-based representations demonstrate poorer performance, even in structure-based splits. This suggests that structural representations may not encode the high-level features necessary for predicting functional characteristics, instead being more specialized in capturing structural details. In contrast, protein language models have been shown to effectively capture both rich functional and structural features (Rives et al., 2021; Lin et al., 2022).

## 7 CONCLUSION

We introduce a fast and methodology for constructing leakage-free, well-balanced, and sufficiently separated data splits. Our approach involves two main steps: first, identifying densely connected nodes within a dataset graph; second, strategically removing hub nodes to fragment the evolutionary structure of the data space. While applied here to proteins, this methodology is data-agnostic and can be extended to other data types, such as DNA/RNA or small molecules, provided a similarity matrix can be computed. We demonstrate its effectiveness by evaluating learned protein representations for function prediction, revealing that structure-based splits present a significantly more challenging test than sequence-based splits, and highlighting that protein function prediction remains an unsolved task. Our source code and data splits are available at a public GitHub repository.

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

# A   DATA LEAKAGE IN THE DEEPFRI SPLITS

Here, we show how more permissive clustering approaches can cause significant data leakage between data splits. We estimate data leakage between the training and validation sets in the DeepFRI splits, which were constructed with CD-HIT clustering. For each sequence identity cutoff, we identify the proteins in the test set that are more similar to proteins in the training set than the identity cutoff. We call these proteins **leaky**. Figure 5 illustrates the proportion of leaky proteins in the test set across different cutoff identity thresholds. Data leakage is substantial at all thresholds, with particularly high levels at lower sequence identities. Additionally, we show the average similarity between these leaky proteins and the training data in Figure 6. We find that the average sequence identity of leaky protein pairs can significantly exceed the specified cutoff threshold. For example, at the 50% sequence identity cutoff, the average sequence identity of leaky protein pairs is 68% in the EC dataset and 76% in the GO dataset, surpassing the allowed cutoff by 36% and 52%, respectively.

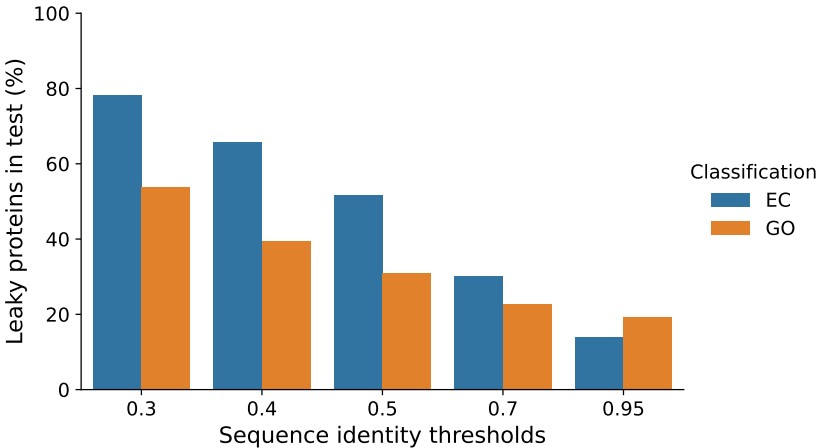

Figure 5: Proportion of proteins in the original DeepFRI test set that are more similar to the training set than the sequence identity cutoff threshold.

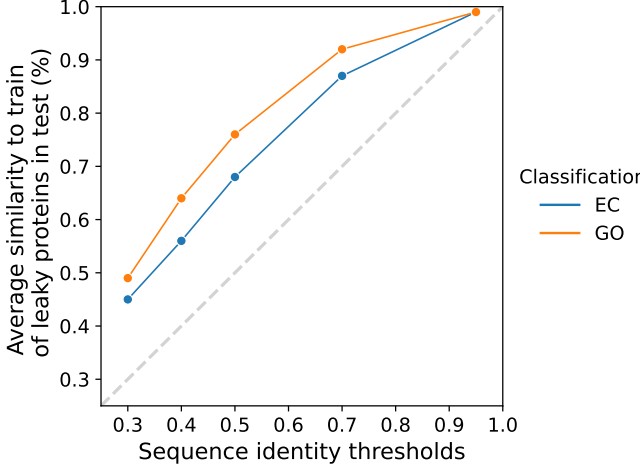

Figure 6: Average sequence identity of leaky protein pairs between the training and test sets in the original DeepFRI splits.

Data leakage may arise from the sequence alignment process used by CD-HIT, which might be less effective than MMseqs2. However, we believe that much of the leakage results from the posterior clustering stage, which may prioritize increasing the number of clusters at the expense of their separation. Similarly, in MMseqs2, the default clustering mode groups remote homologs together

(the connected component mode), likely due to its tendency to produce too few and overly large clusters.

# B EFFECT OF REMOVING CENTRAL POINTS ON THE CLUSTER DISTRIBUTION

**Disconnecting the protein space.**    Figures 8 and 7 illustrate how our community-based clustering approach efficiently fragments the protein space into homogeneously sized components at various similarity thresholds by removing highly central points. In our experiments, we set the resolution parameter $r$ for community detection to 2 (refer to the Leiden algorithm in Algorithm 1), as this value provides the optimal balance between generating more clusters and minimizing protein removal.

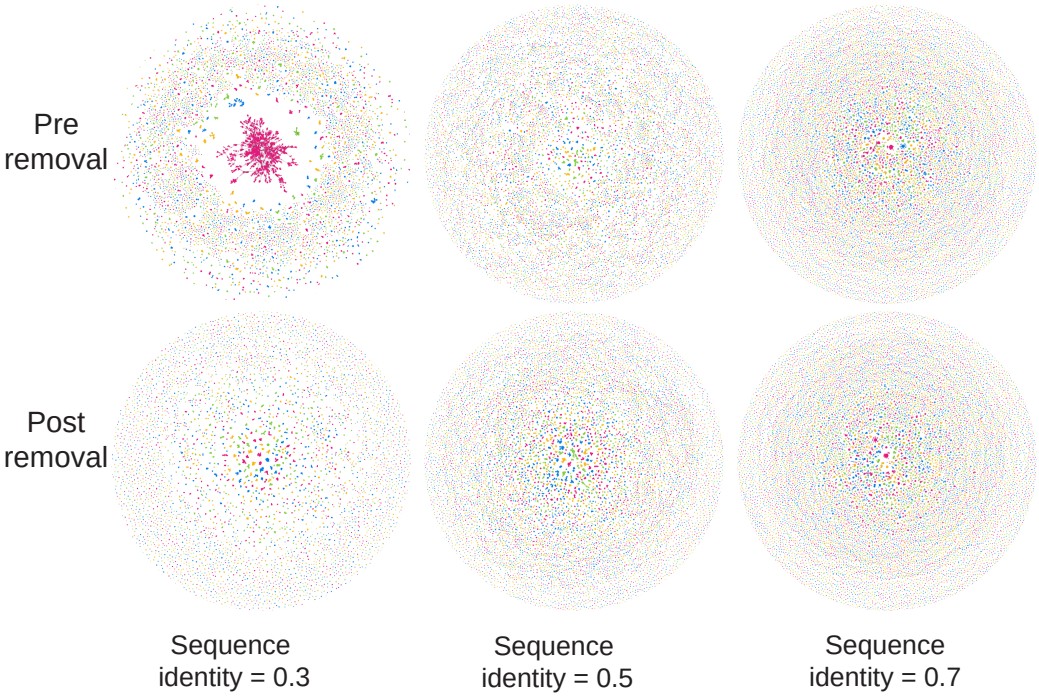

Figure 7: Graph visualization of the EC dataset at increasing sequence identity cutoff thresholds, pre (top row) and post (bottom row) central points removal. Each node represents a protein, and two proteins are connected if their similarity score is superior to the given sequence identity cutoff.

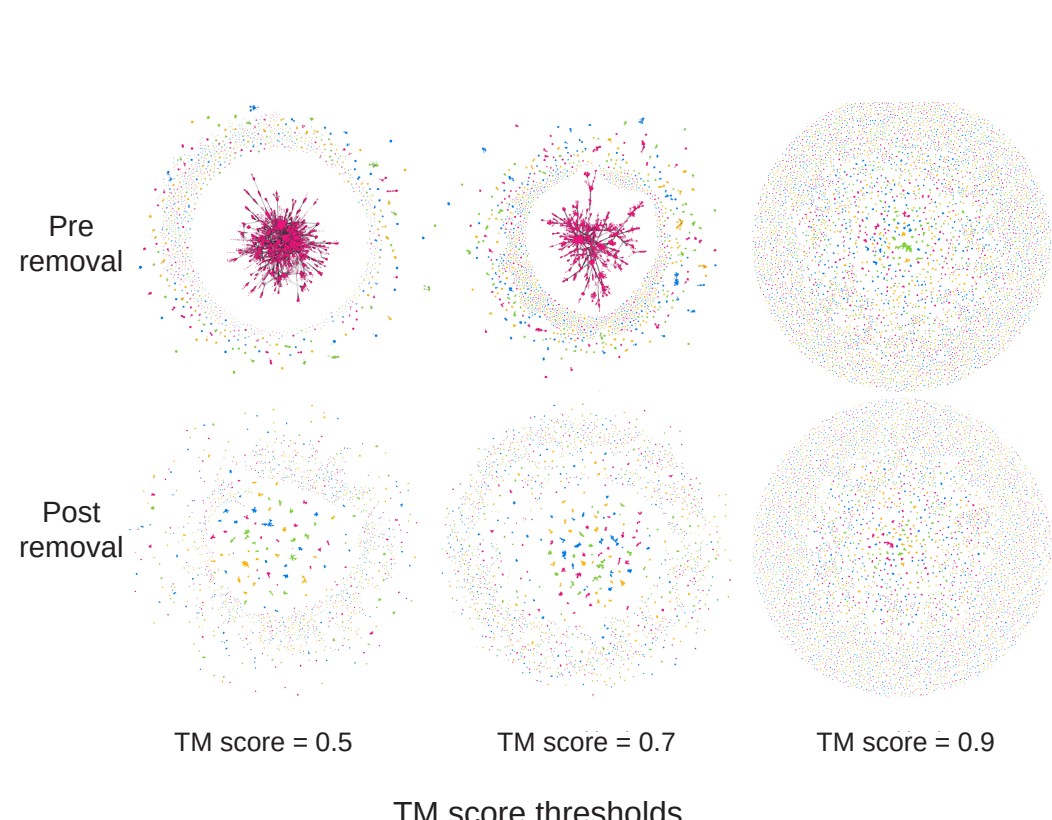

Pre removal

Post removal

TM score = 0.5    TM score = 0.7    TM score = 0.9

TM score thresholds

Figure 8: Graph visualization of the EC dataset at increasing TM score cutoff thresholds, pre (top row) and post (bottom row) central points removal. Each node represents a protein, and two proteins are connected if their similarity score is superior to given TM score cutoff.

**Cluster distributions.** Here, we provide a quantitative visualization of the distributions of connected components before and after removing top connectors from the data. Figure 9 illustrates how the largest components represent a much smaller proportion of the dataset following the community disconnection step. Figure 10 displays the sorted distribution of cluster sizes on a logarithmic scale before and after this process.

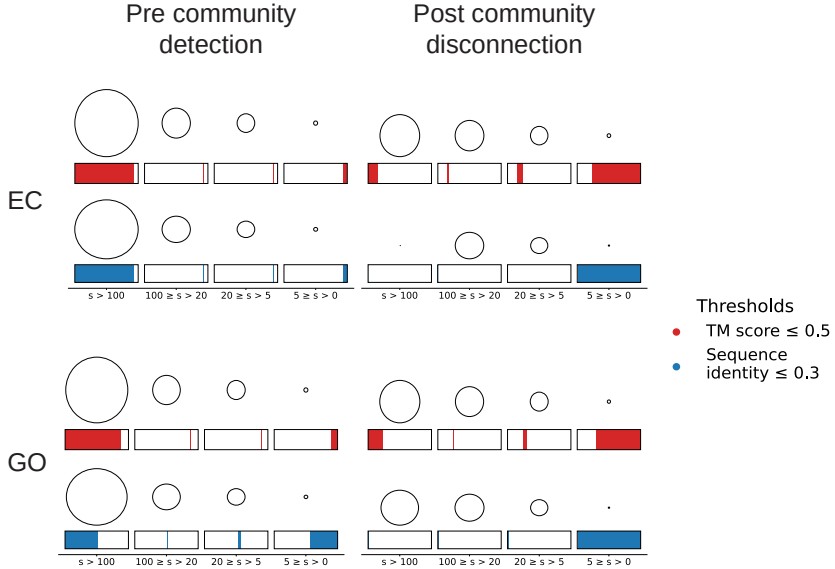

Figure 9: Effect of removing central points on cluster proportions in the EC and GO datasets. Colored rectangles correspond to the proportion that clusters of a certain size $s$ represent in the overall dataset. Circles are proportional to the average cluster size.

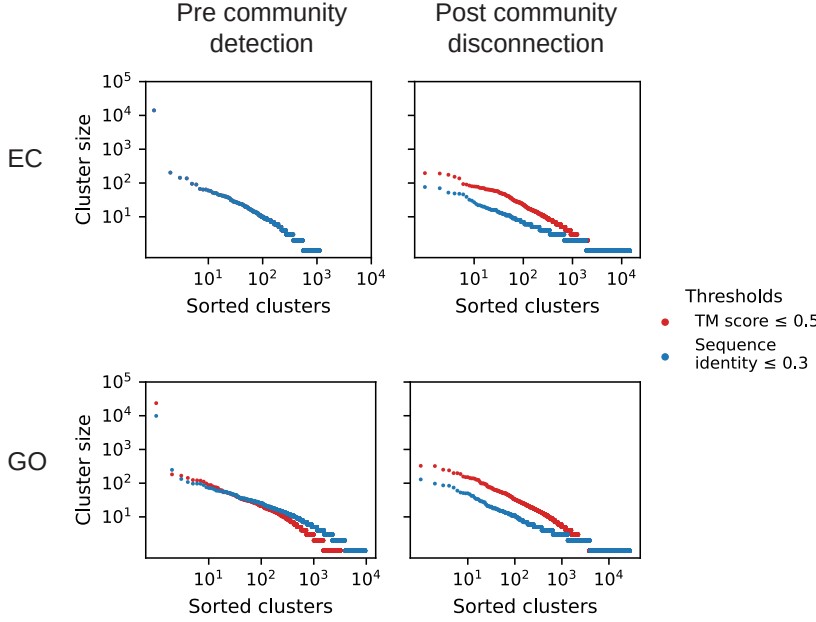

Figure 10: Effect of removing central points on cluster sizes in the EC and GO datasets.

## C EFFECT OF REMOVING CENTRAL POINTS ON MODEL PERFORMANCE

In Figure 11, we compare model performance across identical sequence identity thresholds for the DeepFRI splits and our splits. We observe that performance is more sensitive to similarity thresholds in our splits, which is desirable for evaluating model generalizability. In contrast, the DeepFRI splits show closer results across similarity thresholds, indicating the presence of similar, leaky proteins between the test subsets at different difficulty levels.

Additionally, we note that performance on the EC dataset is higher in the DeepFRI splits compared to our splits, which is expected due to leakage between the training and validation sets. We observe the opposite trend for the GO dataset (BP, CC, and MF), likely due to the randomness of the splitting strategy. Specifically, the DeepFRI split for the GO dataset seems unusually challenging, which accounts for the lower performance. This effect might have been mitigated by averaging over multiple splits, but unlike our approach, DeepFRI used only a single data split.

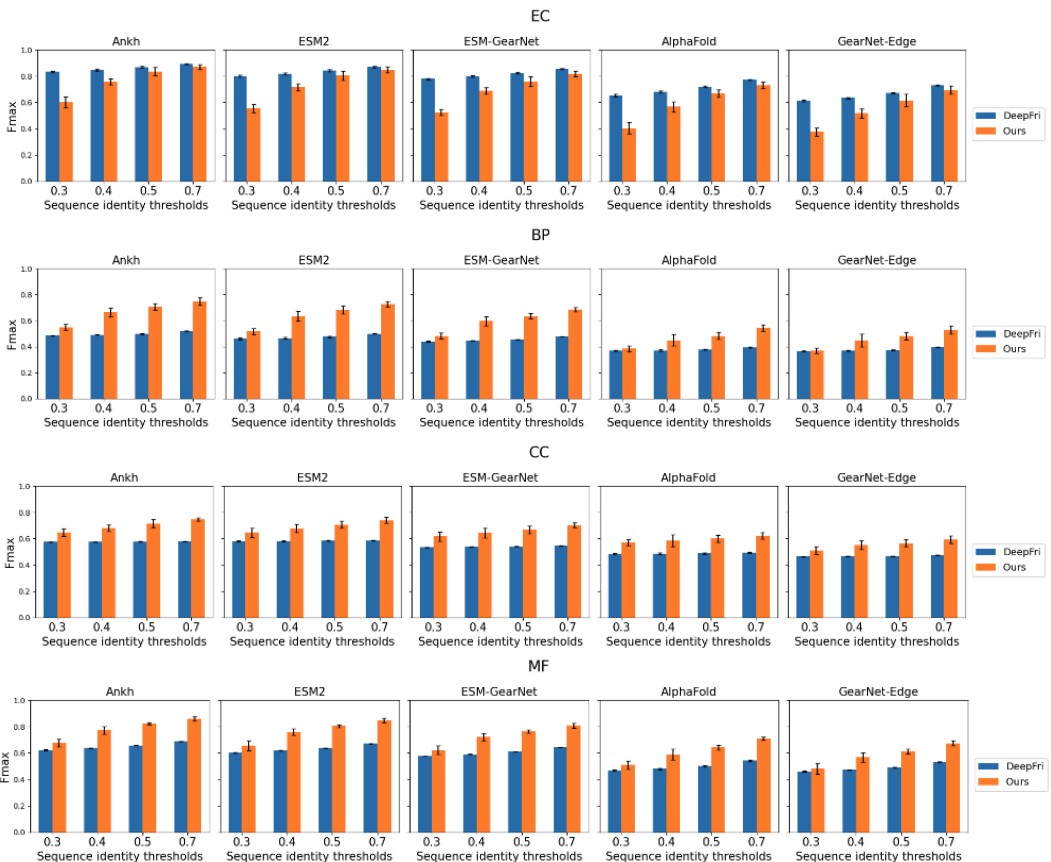

Figure 11: Comparison of model performance on DeepFRI splits and our splits.

## D SEQUENCE VS STRUCTURE SIMILARITY

We hypothesized that i) generalizing to structures would be more challenging than to sequences, and ii) that existing sequence-based splits may not accurately reflect performance on structure-based splits, as proteins with high structural similarity can exhibit significantly different sequences. Figure 12 illustrates this with an example from the EC dataset.

To assess the correlation between sequence identity and structural similarity, Figure 13 presents a density plot of the distributions of TM scores and sequence identities for all protein pairs in the EC dataset. Interestingly, many protein pairs exhibiting high structural similarity (TM score around 0.9) have low sequence identity (around 30%). In contrast, there are very few pairs with highly

A.

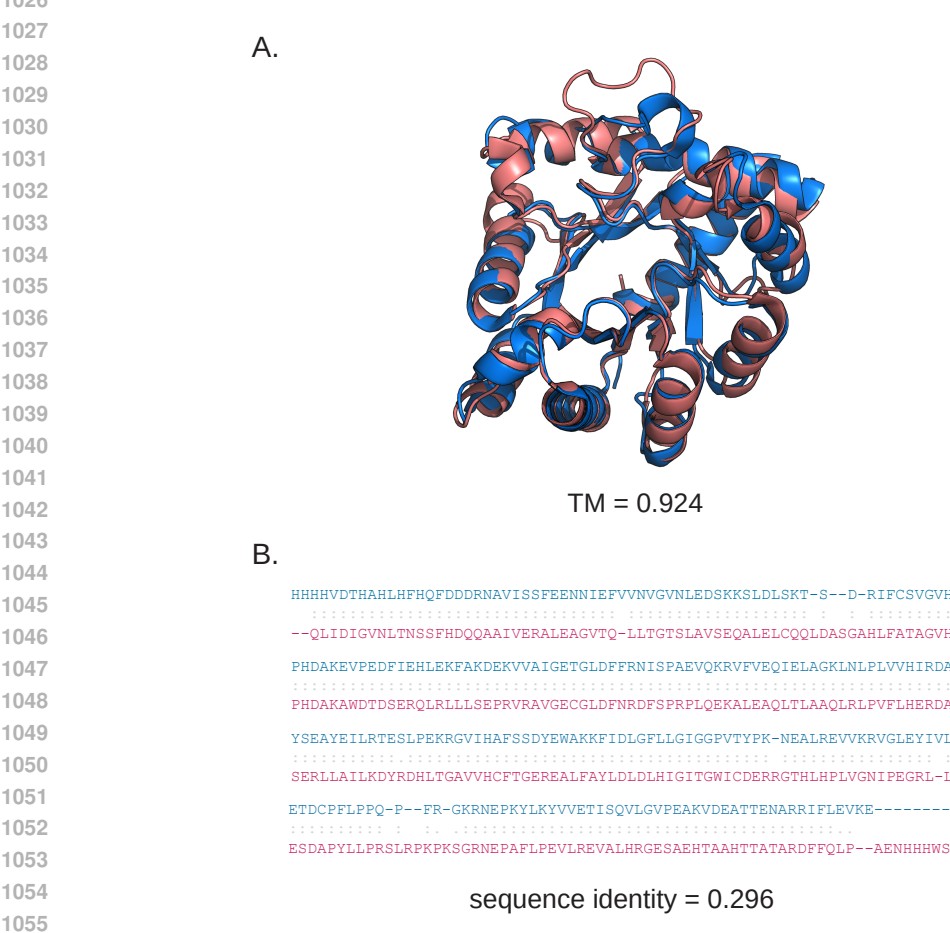

TM = 0.924

B.

```
HHHHVDTHAHLHFHQFDDDRNAVISSFEENNIEFVVNVGVNLEDSKKSLDLSKT-S--D-RIFCSVGVH
::::::::::::::::::::::::::::::::: :::::::::::::::::: :  : :::::::::
--QLIDIGVNLTNSSFHDQQAAIVERALEAGVTQ-LLTGTSLAVSEQALELCQQLDASGAHLFATAGVH

PHDAKEVPEDFIEHLEKFAKDEKVVAIGETGLDFFRNISPAEVQKRVFVEQIELAGKLNLPLVVHIRDA
:::::::::::::::::::::::::::::::::::::::::::::::::::::::::::::::::::::::
PHDAKAWDTDSERQLRLLLSEPRVRAVGECGLDFNRDFSPRPLQEKALEAQLTLAAQLRLPVFLHERDA

YSEAYEILRTESLPEKRGVIHAFSSDYEWAKKFIDLGFLLGIGGPVTYPK-NEALREVVKRVGLEYIVL
:::::::::::::::::::::::::::::::::::::::::::::::::::: :::::::::::::: :
SERLLAILKDYRDHLTGAVVHCFTGEREALFAYLDLDLHIGITGWICDERRGTHLHPLVGNIPEGRL-L

ETDCPFLPPQ-P--FR-GKRNEPKYLKYVVETISQVLGVPEAKVDEATTENARRIFLEVKE-------
:::::::::: :  :. .::::::::::::::::::::::::::::::::::::::::::::..
ESDAPYLLPRSLRPKPKSGRNEPAFLPEVLREVALHRGESAEHTAAHTTATARDFFQLP--AENHHHWS
```

sequence identity = 0.296

Figure 12: Sequence and structural alignments of two enzymes with highly similar structures yet dissimilar sequence. Red is for 3RCM-A, a TatD family hydrolase, and blue is for 1J6O-A, a TatD-related deoxyribonuclease. (A) Structural alignment, computed with TMAlign. (B) Sequence alignment, computed with MMseqs2.

dissimilar structures (TM score around 0.5) that share high sequence identity. This finding suggests that the TM score may be a more suitable metric for creating challenging train/test splits, aligning with recommendations from (Bushuiev et al., 2024).

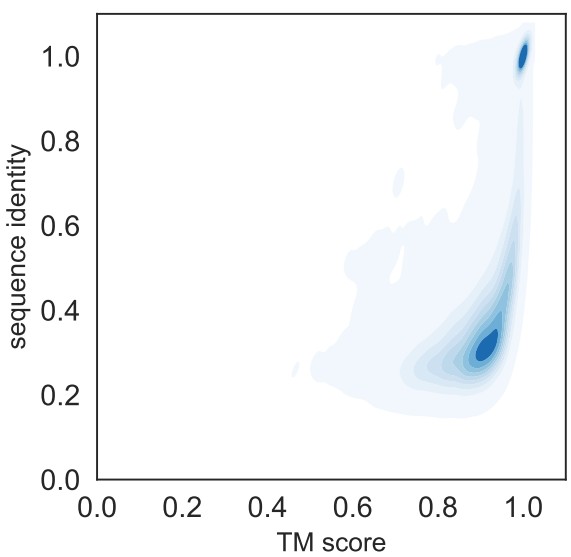

Figure 13: Density plot of TM scores and sequence identities for all protein pairs in the EC dataset. Sequence identities and TM scores were calculated using MMseqs2 and Foldseek, respectively, with a minimum sequence coverage and structure coverage of 80%.

# E    COMPLEXITY ANALYSIS

We estimate the complexity of our method by breaking it down into two main steps:

- **Detecting communities**: We use the Leiden algorithm, which is known for efficiently detecting high-quality communities in large networks. It has been demonstrated to cluster a graph with 10 million nodes and 200 million edges in just over three minutes, with a near-linear runtime in the number of edges. For sparse graphs like those considered here, the number of edges is approximately proportional to the number of nodes, allowing us to estimate the complexity as $O(n \log n)$, where n is the number of nodes.

- **Disconnecting communities**: In this step, the algorithm processes each community (up to n communities). For any given community C, the complexity primarily comes from removing the inter-community edges caused by the removal of the top connector v in C. The number of such connections is at most n. Thus, the overall complexity for this step is upper-bounded by $O(n^2)$.

Combined, the overall complexity of our method is estimated to be quadratic in the number of samples, or $O(n^2)$.

# F    ADDITIONAL DETAILS ON THE EXPERIMENTAL SETUP

## F.1    METRICS

We compute the protein-centric $F_{max}$ score and label-centric AUPRC score according to the definitions provided by CAFA.

**$F_{max}$ score:**    Protein-centric $F_{max}$ corresponds to the maximum $F_1$ score over thresholds $t \in [0, 1]$:

$$F_{max} = \max_t \left( 2 \frac{p(t) \cdot r(t)}{p(t) + r(t)} \right) \tag{1}$$

where, for any given threshold $t$ and protein $i$, precision $p(t)$ is averaged over all proteins, $M(t)$, with at least one positive label, and recall $r(t)$ is averaged over the total number of proteins $N$:

$$\begin{cases} p(t) = \frac{1}{M(t)} \sum_i p_i(t) \\ \\ r(t) = \frac{1}{N} \sum_i r_i(t) \end{cases} \tag{2}$$

and

$$M(t) = \sum_i \mathbb{1}[\sum_l \mathbb{1}[l \in P_i(t)] > 0] \tag{3}$$

where $l$ is a label, $P_i(t)$ is the set of labels predicted as positive for protein $i$ at threshold $t$, and $\mathbb{1}$ is the indicator function.

For any given protein $i$ at threshold $t$, precision $p_i(t)$ and recall $r_i(t)$ are given by

$$\begin{cases} p_i(t) = \frac{\sum_l \mathbb{1}[l \in P_i(t) \cap T_i]}{\sum_l \mathbb{1}[l \in P_i(t)]} \\ \\ r_i(t) = \frac{\sum_l \mathbb{1}[l \in P_i(t) \cap T_i]}{\sum_l \mathbb{1}[l \in T_i]} \end{cases} \tag{4}$$

where $T_i$ is the set of positive labels for protein $i$.

**AUPRC score:** Label-centric AUPRC, the area under the precision-recall curve, is obtained by calculating precision $p(t)$ and recall $r(t)$ over labels $l$ for all thresholds $t$:

$$
\begin{aligned}
AUPRC \quad &= AUC_t\left[p(t),\ r(t)\right] \\
&= \sum_{k=0}^{K-2}\left(r(t_{k+1}) - r(t_k)\right)\cdot p(t_{k+1})
\end{aligned}
\tag{5}
$$

Precision and recall are calculated following Equation 2, with proteins $i$ replaced by labels $l$:

$$
\begin{cases}
p(t) = \frac{1}{M(t)}\sum_l p_l(t) \\[2mm]
r(t) = \frac{1}{L}\sum_l r_l(t)
\end{cases}
\tag{6}
$$

where $L$ is the total number of labels, and

$$
M(t) = \sum_l \mathbb{1}[\sum_i \mathbb{1}[i \in P_l(t)] > 0]
\tag{7}
$$

Similarly as in Equation 4 (with proteins $i$ replaced by labels $l$):

$$
\begin{cases}
p_l(t) = \frac{\sum_i \mathbb{1}[i \in P_l(t) \cap T_l]}{\sum_i \mathbb{1}[i \in P_l(t)]} \\[3mm]
r_l(t) = \frac{\sum_i \mathbb{1}[i \in P_l(t) \cap T_l]}{\sum_i \mathbb{1}[i \in T_l]}
\end{cases}
\tag{8}
$$

**$F_{max}$-cluster score:** We introduce $F_{max}$-cluster, a variant of the $F_{max}$ score designed to account for cluster size imbalance. Instead of being averaged across proteins, precision and recall are first separately calculated for each cluster $c \in C$, then averaged across clusters:

$$
\begin{cases}
p(t) = \frac{1}{M(t)}\sum_{c \in \mathcal{C}} p_c(t) \\[2mm]
r(t) = \frac{1}{|\mathcal{C}|}\sum_{c \in \mathcal{C}} r_c(t)
\end{cases}
\tag{9}
$$

where precision $p_c(t)$ and recall $r_c(t)$ are calculated across proteins $i$ in cluster $c$:

$$
\begin{cases}
p_c(t) = \frac{1}{M_c(t)}\sum_{i \in c} p_i(t) \\[2mm]
r_c(t) = \frac{1}{N_c}\sum_{i \in c} r_i(t)
\end{cases}
\tag{10}
$$

with

$$
M_c(t) = \sum_{i \in c} \mathbb{1}[\sum_l \mathbb{1}[l \in P_i(t)] > 0]
\tag{11}
$$

### F.2 Input protein representations

We probe representations extracted from the following sequence and structure-based encoders.

**ESM2-650M (Lin et al., 2022)** ESM2-650M is a popular protein language model based on the transformer encoder (Vaswani et al., 2017) and trained with the BERT masked language modeling objective (Devlin et al., 2018). We extract the residue-level representation from the last layer.

**Ankh (Elnaggar et al., 2023)** Ankh, another masked protein language model, is based on the T5 architecture (Raffel et al., 2020). In contrast to the ESM2 model series, Ankh places less emphasis on model scaling and instead focuses on carefully tuning its hyperparameters. As with ESM2, we extract the residue-level representation from the last layer.

**AlphaFold2 (Jumper et al., 2021)** AlphaFold2 is a state-of-the art protein structure prediction model based on two intertwined blocks of transformers. The first block, called the Evoformer, learns dependencies between residues by processing raw multiple sequence alignments. The Evoformer returns an abstract representation which the second transformer block, the Structure module, transforms into residue positions. We extract the protein representation returned by the Evoformer module after it has gone through 3 iteration cycles.

**GearNet-Edge (Zhang et al., 2022)** GearNet-Edge is a graph-based protein structure encoder which encodes spatial information in the edges of protein residue graphs. GearNet-Edge was pre-trained with various self-supervised task. We select the model trained with the multiview contrastive learning objective, as it was reported to outperform other GearNet models on the EC and GO prediction tasks (Zhang et al., 2022) and extract the residue-level representation.

**ESM-GearNet (Zhang et al., 2023)** ESM-GearNet is a hybrid sequence and structure-based encoder, obtained by fusing ESM2 and GearNet. It aims to generate richer protein representations by leveraging signal from both sequence and structure. As previously, we extract the residue-level representation learned by the model.

We emphasize that the goal of this paper is not to outperform baselines on existing benchmarks, but to study the behavior of various protein representations derived from sequence and structure under different data splitting regimes. Nevertheless, a simple experiment to enhance performance could involve probing representations extracted from different layers of the aforementioned models. For example, previous studies on BERT language models indicate that the last layer does not consistently yield superior predictive performance (Rogers et al., 2021).

### F.3 MODEL HYPERPARAMETERS

We train simple, fully-connected neural networks on the protein function prediction tasks. The models can be described as follows: $Output = W_{out} \cdot Dropout(ReLU(W_h \cdot x + b_h) + b_{out}, p)$, where $W_h$ is the weight matrix of the hidden layer, $W_{out}$ is the weight matrix of the output layer, $b_h$ and $b_{out}$ are the biases of the hidden and output layers, respectively, ReLU is the activation function, and dropout is applied with probability $p$ to the activation output. The output layer has $n_{class}$ dimension, where $n_{class}$ is the size of the label one-hot vector for the given ontology.

Table summarizes the set of hyperparameters we select for each protein representation after tuning models on the original DeepFRI splits. Choosing the appropriate combination of batch size and learning rate is critical for successful training. Additionally, a moderate level of dropout proves beneficial. Regarding model size, we opt for the minimum hidden dimension where performance plateaus. Our experiments show that larger hidden dimensions enable faster training but do not necessarily improve performance, and come with increased memory requirements.

Table 1: Model hyperparameters

| DESCRIPTION / REPRESENTATION | ANKH | ESM2-650M | ESM-GEARNET | GEARNET-EDGE | ALPHAFOLD |
|---|---|---|---|---|---|
| DIMENSION OF REPRESENTATION | 1536 | 1280 | 4352 | 3072 | 384 |
| HIDDEN LAYER DIMENSION | 1536 | 1280 | 1024 | 1024 | 1536 |
| LEARNING RATE | 0.003 | 0.003 | 0.001 | 0.0001 | 0.0005 |
| DROPOUT | 0.3 | 0.3 | 0.3 | 0.3 | 0.3 |
| BATCH SIZE | 128 | 128 | 128 | 128 | 128 |

# G   ADDITIONAL RESULTS ON EC AND GO PREDICTION

In this section, we present more results for the EC and GO prediction tasks. To facilitate future benchmarking efforts, results for all models and data splits can be found in Tables 2-13.

**Results on the new splits.**    Similarly to Figure 4 illustrates how protein representations struggle to generalize on structure-based splits compared to sequence-based splits. The results appear slightly inflated, and the variance is more pronounced when evaluating performance using the $F_{max}$ (averaged over proteins) compared to the $F_{max}$-cluster (averaged over protein clusters). This emphasizes the value of a cluster-based metric in performance assessment. The trends observed across splits and similarity thresholds are consistent for the label-centric AUPRC score and are even more pronounced.

**Results on the original DeepFRI splits.**    Figure 15 shows the $F_{max}$ and AUPRC scores for the original, sequence-based DeepFRI splits. The increase in performance between lower and higher similarity thresholds is less pronounced than in our own sequence-based splits. We identify several potential reasons behind this difference, all related to the methodology employed to build the test set.

First, as detailed in A, the DeepFRI splits exhibit significant leakage between the training and test sets. Second, in the DeepFRI splits, proteins from low sequence identity subsets are included in subsets at higher thresholds, meaning that the test subsets are not independent and that performance at any given similarity cutoff will closely resemble performance at the preceding cutoff. Additionally, the number of proteins added to the test set varies across sequence identity cutoffs, leading to inconsistent representation of similarity thresholds in the final results. Overall, these issues hinder fair comparisons between sequence identity thresholds in the DeepFRI splits and may account for the comparable performance observed across thresholds compared to our splits. Note that we do not report results using the $F_{max}$-cluster for the original DeepFRI splits, as cluster information is not available in the dataset.

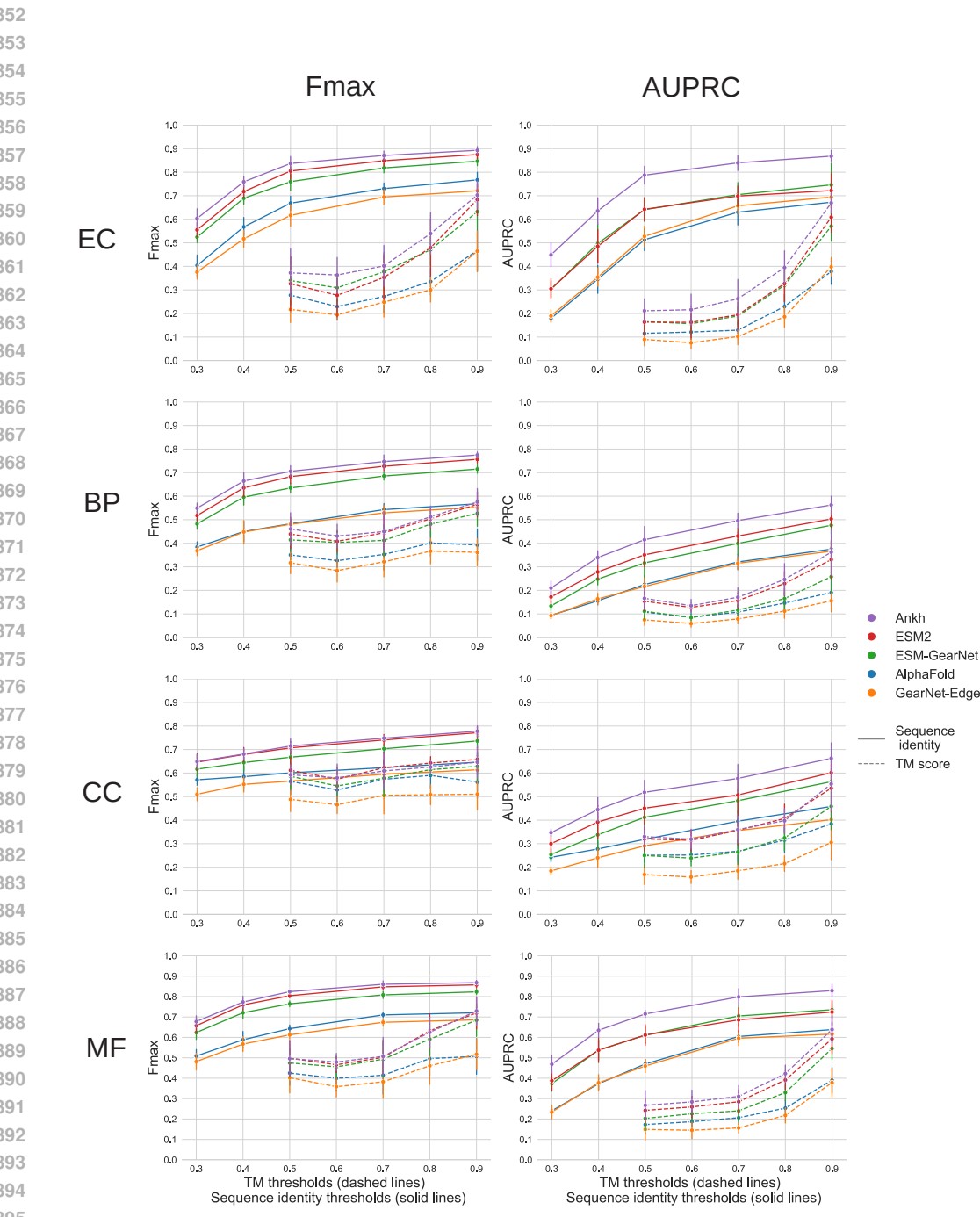

Figure 14: F$_{max}$ and AUPRC on EC and GO prediction. Models are separately trained to predict the three ontologies of the GO dataset: Biological Process (BP), Cellular Component (CC), and Molecular Function (MF). Results on the structure-based splits are shown with dashed lines (TM thresholds between 0.5 and 0.9); results on the sequence-based with solid lines (sequence identity thresholds between 0.3 and 0.9).

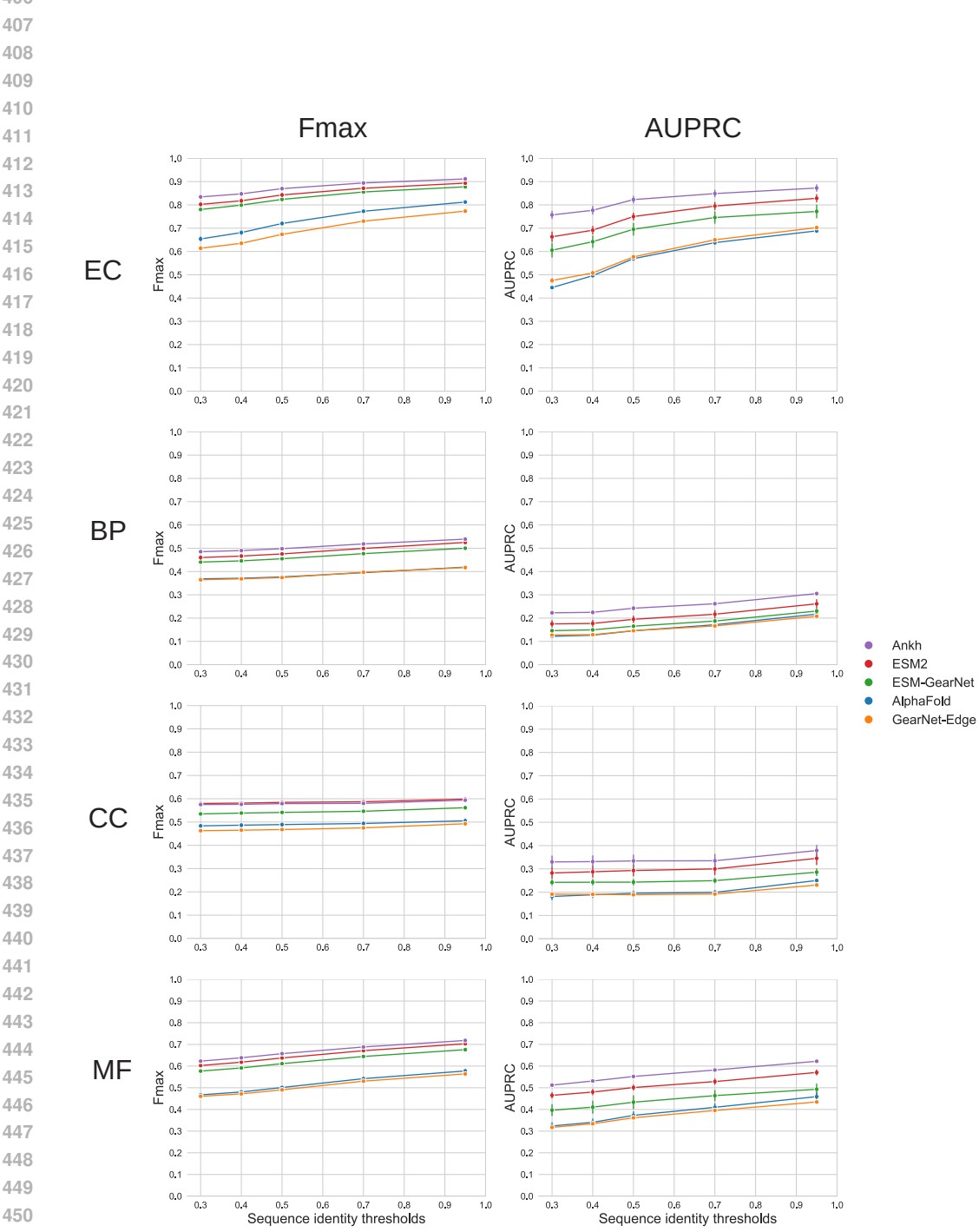

Figure 15: $F_{max}$ and AUPRC on the original, sequence identity-based DeepFRI splits.

Table 2: $F_{max}$-cluster on EC prediction.

| | SEQUENCE IDENTITY | | | | | TM SCORE | | | | |
|---|---|---|---|---|---|---|---|---|---|---|
| MODELS | 0.3 | 0.4 | 0.5 | 0.7 | 0.9 | 0.5 | 0.6 | 0.7 | 0.8 | 0.9 |
| ANKH | 0.15 | 0.23 | 0.28 | 0.76 | 0.36 | 0.31 | 0.37 | 0.5 | 0.55 | 0.59 |
| ESM | 0.14 | 0.2 | 0.29 | 0.78 | 0.36 | 0.43 | 0.48 | 0.63 | 0.67 | 0.71 |
| ESM-GEARNET | 0.14 | 0.22 | 0.28 | 0.81 | 0.38 | 0.54 | 0.6 | 0.71 | 0.76 | 0.8 |
| AF-SEVO | 0.22 | 0.29 | 0.4 | 0.84 | 0.49 | 0.62 | 0.67 | 0.78 | 0.81 | 0.83 |
| GEARNET-EDGE | 0.39 | 0.39 | 0.61 | 0.87 | 0.69 | 0.67 | 0.72 | 0.81 | 0.84 | 0.86 |

Table 3: $F_{max}$ on EC prediction.

| | DEEPFRI | | | | | SEQUENCE IDENTITY | | | | | TM SCORE | | | | |
|---|---|---|---|---|---|---|---|---|---|---|---|---|---|---|---|
| MODELS | 0.3 | 0.4 | 0.5 | 0.7 | 0.95 | 0.3 | 0.4 | 0.5 | 0.7 | 0.9 | 0.5 | 0.6 | 0.7 | 0.8 | 0.9 |
| ANKH | 0.22 | 0.28 | 0.22 | 0.33 | 0.37 | 0.38 | 0.4 | 0.52 | 0.56 | 0.6 | 0.61 | 0.65 | 0.78 | 0.8 | 0.83 |
| ESM | 0.19 | 0.23 | 0.19 | 0.28 | 0.36 | 0.52 | 0.57 | 0.69 | 0.72 | 0.76 | 0.64 | 0.68 | 0.8 | 0.82 | 0.85 |
| ESM-GEARNET | 0.25 | 0.27 | 0.25 | 0.35 | 0.4 | 0.62 | 0.67 | 0.76 | 0.81 | 0.84 | 0.67 | 0.72 | 0.82 | 0.84 | 0.87 |
| AF-SEVO | 0.3 | 0.34 | 0.3 | 0.48 | 0.54 | 0.69 | 0.73 | 0.82 | 0.85 | 0.87 | 0.73 | 0.77 | 0.86 | 0.87 | 0.89 |
| GEARNET-EDGE | 0.46 | 0.47 | 0.46 | 0.68 | 0.7 | 0.72 | 0.77 | 0.85 | 0.88 | 0.89 | 0.77 | 0.81 | 0.88 | 0.89 | 0.91 |

Table 4: AUPRC on EC prediction.

| | DEEPFRI | | | | | SEQUENCE IDENTITY | | | | | TM SCORE | | | | |
|---|---|---|---|---|---|---|---|---|---|---|---|---|---|---|---|
| MODELS | 0.3 | 0.4 | 0.5 | 0.7 | 0.95 | 0.3 | 0.4 | 0.5 | 0.7 | 0.9 | 0.5 | 0.6 | 0.7 | 0.8 | 0.9 |
| ANKH | 0.09 | 0.12 | 0.09 | 0.16 | 0.21 | 0.19 | 0.18 | 0.3 | 0.3 | 0.45 | 0.48 | 0.44 | 0.61 | 0.66 | 0.76 |
| ESM | 0.08 | 0.12 | 0.08 | 0.16 | 0.22 | 0.35 | 0.35 | 0.5 | 0.48 | 0.64 | 0.51 | 0.5 | 0.64 | 0.69 | 0.78 |
| ESM-GEARNET | 0.1 | 0.13 | 0.1 | 0.19 | 0.26 | 0.53 | 0.51 | 0.64 | 0.64 | 0.79 | 0.58 | 0.57 | 0.7 | 0.75 | 0.82 |
| AF-SEVO | 0.19 | 0.23 | 0.19 | 0.33 | 0.4 | 0.66 | 0.63 | 0.7 | 0.7 | 0.84 | 0.65 | 0.64 | 0.75 | 0.79 | 0.85 |
| GEARNET-EDGE | 0.4 | 0.38 | 0.4 | 0.61 | 0.67 | 0.69 | 0.67 | 0.75 | 0.72 | 0.87 | 0.7 | 0.69 | 0.77 | 0.83 | 0.87 |

Table 5: $F_{max}$-cluster on BP prediction.

| MODELS | SEQUENCE IDENTITY | | | | | TM SCORE | | | | |
|---|---|---|---|---|---|---|---|---|---|---|
| | 0.3 | 0.4 | 0.5 | 0.7 | 0.9 | 0.5 | 0.6 | 0.7 | 0.8 | 0.9 |
| ANKH | 0.15 | 0.23 | 0.28 | 0.76 | 0.36 | 0.31 | 0.37 | 0.5 | 0.55 | 0.59 |
| ESM | 0.14 | 0.2 | 0.29 | 0.78 | 0.36 | 0.43 | 0.48 | 0.63 | 0.67 | 0.71 |
| ESM-GEARNET | 0.14 | 0.22 | 0.28 | 0.81 | 0.38 | 0.54 | 0.6 | 0.71 | 0.76 | 0.8 |
| AF-SEVO | 0.22 | 0.29 | 0.4 | 0.84 | 0.49 | 0.62 | 0.67 | 0.78 | 0.81 | 0.83 |
| GEARNET-EDGE | 0.39 | 0.39 | 0.61 | 0.87 | 0.69 | 0.67 | 0.72 | 0.81 | 0.84 | 0.86 |

Table 6: $F_{max}$ on BP prediction.

| MODELS | DEEPFRI | | | | | SEQUENCE IDENTITY | | | | | TM SCORE | | | | |
|---|---|---|---|---|---|---|---|---|---|---|---|---|---|---|---|
| | 0.3 | 0.4 | 0.5 | 0.7 | 0.95 | 0.3 | 0.4 | 0.5 | 0.7 | 0.9 | 0.5 | 0.6 | 0.7 | 0.8 | 0.9 |
| ANKH | 0.32 | 0.35 | 0.32 | 0.44 | 0.46 | 0.37 | 0.38 | 0.48 | 0.52 | 0.55 | 0.36 | 0.37 | 0.44 | 0.46 | 0.49 |
| ESM | 0.28 | 0.33 | 0.28 | 0.41 | 0.43 | 0.45 | 0.45 | 0.6 | 0.64 | 0.66 | 0.37 | 0.37 | 0.45 | 0.47 | 0.49 |
| ESM-GEARNET | 0.32 | 0.35 | 0.32 | 0.44 | 0.45 | 0.48 | 0.48 | 0.63 | 0.68 | 0.71 | 0.37 | 0.38 | 0.45 | 0.48 | 0.5 |
| AF-SEVO | 0.37 | 0.4 | 0.37 | 0.5 | 0.51 | 0.53 | 0.54 | 0.69 | 0.73 | 0.75 | 0.4 | 0.4 | 0.48 | 0.5 | 0.52 |
| GEARNET-EDGE | 0.36 | 0.39 | 0.36 | 0.57 | 0.58 | 0.55 | 0.57 | 0.72 | 0.76 | 0.77 | 0.42 | 0.42 | 0.5 | 0.52 | 0.54 |

Table 7: AUPRC on BP prediction.

| MODELS | DEEPFRI | | | | | SEQUENCE IDENTITY | | | | | TM SCORE | | | | |
|---|---|---|---|---|---|---|---|---|---|---|---|---|---|---|---|
| | 0.3 | 0.4 | 0.5 | 0.7 | 0.95 | 0.3 | 0.4 | 0.5 | 0.7 | 0.9 | 0.5 | 0.6 | 0.7 | 0.8 | 0.9 |
| ANKH | 0.07 | 0.11 | 0.07 | 0.15 | 0.17 | 0.09 | 0.09 | 0.13 | 0.17 | 0.21 | 0.13 | 0.12 | 0.15 | 0.18 | 0.22 |
| ESM | 0.06 | 0.09 | 0.06 | 0.13 | 0.13 | 0.16 | 0.16 | 0.25 | 0.28 | 0.34 | 0.13 | 0.13 | 0.15 | 0.18 | 0.22 |
| ESM-GEARNET | 0.08 | 0.11 | 0.08 | 0.16 | 0.17 | 0.22 | 0.22 | 0.32 | 0.35 | 0.41 | 0.15 | 0.15 | 0.17 | 0.19 | 0.24 |
| AF-SEVO | 0.11 | 0.15 | 0.11 | 0.23 | 0.25 | 0.32 | 0.32 | 0.4 | 0.43 | 0.5 | 0.17 | 0.17 | 0.19 | 0.22 | 0.26 |
| GEARNET-EDGE | 0.16 | 0.19 | 0.16 | 0.33 | 0.36 | 0.37 | 0.38 | 0.48 | 0.5 | 0.56 | 0.21 | 0.22 | 0.23 | 0.26 | 0.31 |

Table 8: $F_{max}$-cluster on CC prediction.

| MODELS | SEQUENCE IDENTITY | | | | | TM SCORE | | | | |
|---|---|---|---|---|---|---|---|---|---|---|
| | 0.3 | 0.4 | 0.5 | 0.7 | 0.9 | 0.5 | 0.6 | 0.7 | 0.8 | 0.9 |
| ANKH | 0.15 | 0.23 | 0.28 | 0.76 | 0.36 | 0.31 | 0.37 | 0.5 | 0.55 | 0.59 |
| ESM | 0.14 | 0.2 | 0.29 | 0.78 | 0.36 | 0.43 | 0.48 | 0.63 | 0.67 | 0.71 |
| ESM-GEARNET | 0.14 | 0.22 | 0.28 | 0.81 | 0.38 | 0.54 | 0.6 | 0.71 | 0.76 | 0.8 |
| AF-SEVO | 0.22 | 0.29 | 0.4 | 0.84 | 0.49 | 0.62 | 0.67 | 0.78 | 0.81 | 0.83 |
| GEARNET-EDGE | 0.39 | 0.39 | 0.61 | 0.87 | 0.69 | 0.67 | 0.72 | 0.81 | 0.84 | 0.86 |

Table 9: $F_{max}$ on CC prediction.

| MODELS | DEEPFRI | | | | | SEQUENCE IDENTITY | | | | | TM SCORE | | | | |
|---|---|---|---|---|---|---|---|---|---|---|---|---|---|---|---|
| | 0.3 | 0.4 | 0.5 | 0.7 | 0.95 | 0.3 | 0.4 | 0.5 | 0.7 | 0.9 | 0.5 | 0.6 | 0.7 | 0.8 | 0.9 |
| ANKH | 0.49 | 0.56 | 0.49 | 0.61 | 0.59 | 0.51 | 0.57 | 0.62 | 0.65 | 0.65 | 0.46 | 0.48 | 0.53 | 0.58 | 0.58 |
| ESM | 0.47 | 0.53 | 0.47 | 0.57 | 0.58 | 0.55 | 0.58 | 0.64 | 0.68 | 0.68 | 0.46 | 0.49 | 0.54 | 0.58 | 0.58 |
| ESM-GEARNET | 0.51 | 0.57 | 0.51 | 0.62 | 0.61 | 0.57 | 0.6 | 0.67 | 0.71 | 0.72 | 0.47 | 0.49 | 0.54 | 0.58 | 0.58 |
| AF-SEVO | 0.51 | 0.59 | 0.51 | 0.64 | 0.63 | 0.59 | 0.62 | 0.7 | 0.74 | 0.75 | 0.47 | 0.49 | 0.55 | 0.59 | 0.58 |
| GEARNET-EDGE | 0.51 | 0.56 | 0.51 | 0.66 | 0.65 | 0.61 | 0.65 | 0.74 | 0.77 | 0.78 | 0.49 | 0.51 | 0.56 | 0.6 | 0.59 |

Table 10: AUPRC on CC prediction.

| MODELS | DEEPFRI | | | | | SEQUENCE IDENTITY | | | | | TM SCORE | | | | |
|---|---|---|---|---|---|---|---|---|---|---|---|---|---|---|---|
| | 0.3 | 0.4 | 0.5 | 0.7 | 0.95 | 0.3 | 0.4 | 0.5 | 0.7 | 0.9 | 0.5 | 0.6 | 0.7 | 0.8 | 0.9 |
| ANKH | 0.17 | 0.25 | 0.17 | 0.32 | 0.33 | 0.18 | 0.24 | 0.25 | 0.3 | 0.35 | 0.19 | 0.18 | 0.24 | 0.28 | 0.33 |
| ESM | 0.16 | 0.25 | 0.16 | 0.31 | 0.32 | 0.24 | 0.28 | 0.34 | 0.39 | 0.44 | 0.19 | 0.19 | 0.24 | 0.29 | 0.33 |
| ESM-GEARNET | 0.18 | 0.27 | 0.18 | 0.36 | 0.36 | 0.29 | 0.32 | 0.41 | 0.45 | 0.52 | 0.19 | 0.2 | 0.24 | 0.29 | 0.33 |
| AF-SEVO | 0.22 | 0.32 | 0.22 | 0.41 | 0.4 | 0.36 | 0.4 | 0.48 | 0.51 | 0.58 | 0.19 | 0.2 | 0.25 | 0.3 | 0.33 |
| GEARNET-EDGE | 0.31 | 0.38 | 0.31 | 0.54 | 0.55 | 0.4 | 0.46 | 0.56 | 0.6 | 0.66 | 0.23 | 0.25 | 0.29 | 0.35 | 0.38 |

Table 11: $F_{max}$-cluster on MF prediction.

| MODELS | SEQUENCE IDENTITY | | | | | TM SCORE | | | | |
|---|---|---|---|---|---|---|---|---|---|---|
| | 0.3 | 0.4 | 0.5 | 0.7 | 0.9 | 0.5 | 0.6 | 0.7 | 0.8 | 0.9 |
| ANKH | 0.15 | 0.23 | 0.28 | 0.76 | 0.36 | 0.31 | 0.37 | 0.5 | 0.55 | 0.59 |
| ESM | 0.14 | 0.2 | 0.29 | 0.78 | 0.36 | 0.43 | 0.48 | 0.63 | 0.67 | 0.71 |
| ESM-GEARNET | 0.14 | 0.22 | 0.28 | 0.81 | 0.38 | 0.54 | 0.6 | 0.71 | 0.76 | 0.8 |
| AF-SEVO | 0.22 | 0.29 | 0.4 | 0.84 | 0.49 | 0.62 | 0.67 | 0.78 | 0.81 | 0.83 |
| GEARNET-EDGE | 0.39 | 0.39 | 0.61 | 0.87 | 0.69 | 0.67 | 0.72 | 0.81 | 0.84 | 0.86 |

Table 12: $F_{max}$ on MF prediction.

| MODELS | DEEPFRI | | | | | SEQUENCE IDENTITY | | | | | TM SCORE | | | | |
|---|---|---|---|---|---|---|---|---|---|---|---|---|---|---|---|
| | 0.3 | 0.4 | 0.5 | 0.7 | 0.95 | 0.3 | 0.4 | 0.5 | 0.7 | 0.9 | 0.5 | 0.6 | 0.7 | 0.8 | 0.9 |
| ANKH | 0.4 | 0.43 | 0.4 | 0.5 | 0.5 | 0.48 | 0.51 | 0.62 | 0.66 | 0.68 | 0.46 | 0.47 | 0.58 | 0.6 | 0.62 |
| ESM | 0.36 | 0.4 | 0.36 | 0.47 | 0.48 | 0.57 | 0.59 | 0.72 | 0.76 | 0.77 | 0.47 | 0.48 | 0.59 | 0.62 | 0.64 |
| ESM-GEARNET | 0.38 | 0.41 | 0.38 | 0.5 | 0.51 | 0.61 | 0.64 | 0.76 | 0.8 | 0.82 | 0.49 | 0.5 | 0.61 | 0.64 | 0.66 |
| AF-SEVO | 0.46 | 0.5 | 0.46 | 0.63 | 0.62 | 0.67 | 0.71 | 0.81 | 0.85 | 0.86 | 0.53 | 0.54 | 0.64 | 0.67 | 0.69 |
| GEARNET-EDGE | 0.52 | 0.51 | 0.52 | 0.72 | 0.73 | 0.69 | 0.72 | 0.82 | 0.86 | 0.87 | 0.56 | 0.58 | 0.68 | 0.7 | 0.72 |

Table 13: AUPRC on MF prediction.

| MODELS | DEEPFRI | | | | | SEQUENCE IDENTITY | | | | | TM SCORE | | | | |
|---|---|---|---|---|---|---|---|---|---|---|---|---|---|---|---|
| | 0.3 | 0.4 | 0.5 | 0.7 | 0.95 | 0.3 | 0.4 | 0.5 | 0.7 | 0.9 | 0.5 | 0.6 | 0.7 | 0.8 | 0.9 |
| ANKH | 0.15 | 0.17 | 0.15 | 0.24 | 0.27 | 0.23 | 0.24 | 0.37 | 0.39 | 0.47 | 0.32 | 0.32 | 0.4 | 0.46 | 0.51 |
| ESM | 0.14 | 0.19 | 0.14 | 0.26 | 0.28 | 0.38 | 0.37 | 0.54 | 0.54 | 0.63 | 0.33 | 0.34 | 0.41 | 0.48 | 0.53 |
| ESM-GEARNET | 0.16 | 0.21 | 0.16 | 0.28 | 0.31 | 0.46 | 0.47 | 0.61 | 0.61 | 0.71 | 0.36 | 0.37 | 0.43 | 0.5 | 0.55 |
| AF-SEVO | 0.22 | 0.25 | 0.22 | 0.39 | 0.42 | 0.6 | 0.6 | 0.7 | 0.69 | 0.8 | 0.4 | 0.41 | 0.46 | 0.53 | 0.58 |
| GEARNET-EDGE | 0.38 | 0.39 | 0.38 | 0.59 | 0.64 | 0.62 | 0.64 | 0.74 | 0.72 | 0.83 | 0.43 | 0.46 | 0.49 | 0.57 | 0.62 |

# H ADDITIONAL CLUSTER-BASED RESULTS

Here, we evaluate model performance on individual clusters in the EC dataset. In contrast to results measured with the $F_{max}$-cluster (Figure 4), we do not report aggregate scores over all protein clusters. Figures 16-18 show that both the $F_{max}$ score (on top) and the AUPRC score (below) exhibit bimodal distributions across all protein representations and similarity thresholds. Models frequently predict individual clusters either very well or very poorly, highlighting varying difficulty levels among protein families. This pattern is particularly evident for clusters with fewer members, which, as shown in Figures 9 and 10, constitute a significant portion of the overall cluster population. The pronounced performance differences likely stem from the distance of some clusters to the training set. For example, certain clusters at the 90% similarity threshold may exhibit a very low similarity (e.g 30%) to the training set, resulting in a decline in performance.

Furthermore, we find no significant difference in performance between sequence-based and structure-based representations.

We also compare the behavior of the $F_{max}$ score and AUPRC score across clusters of varying sizes in Figure 19. Interestingly, while the Fmax score remains relatively stable, the AUPRC score shows a negative correlation with cluster size. This suggests that the AUPRC may not be the most suitable metric for evaluating model performance on individual clusters.

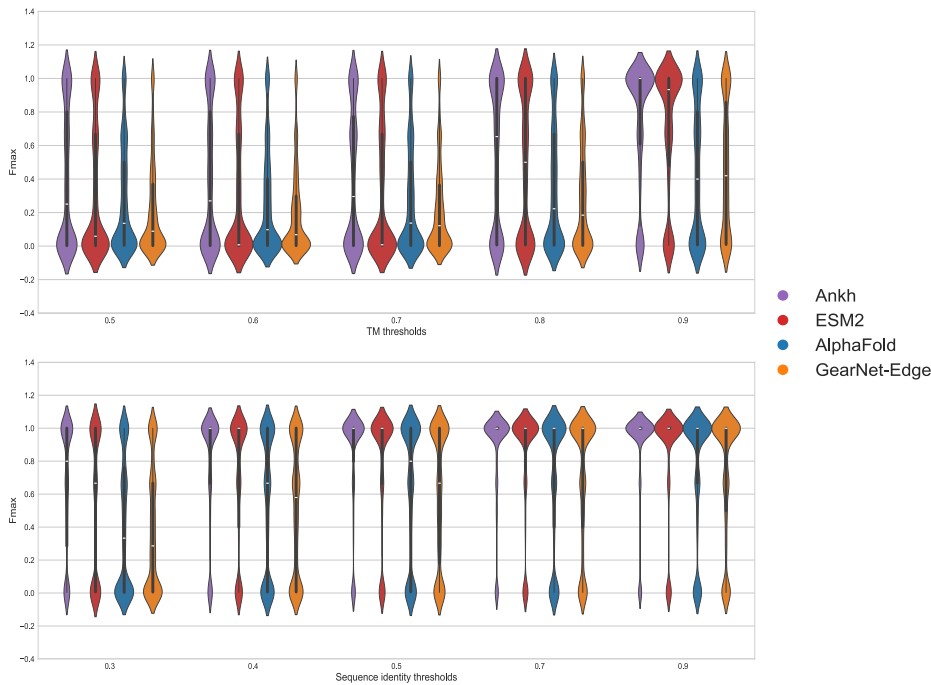

Figure 16: Distributions of $F_{max}$ and AUPCR scores for all individual clusters in the EC prediction task. Results are aggregated across all models and dataset seeds.

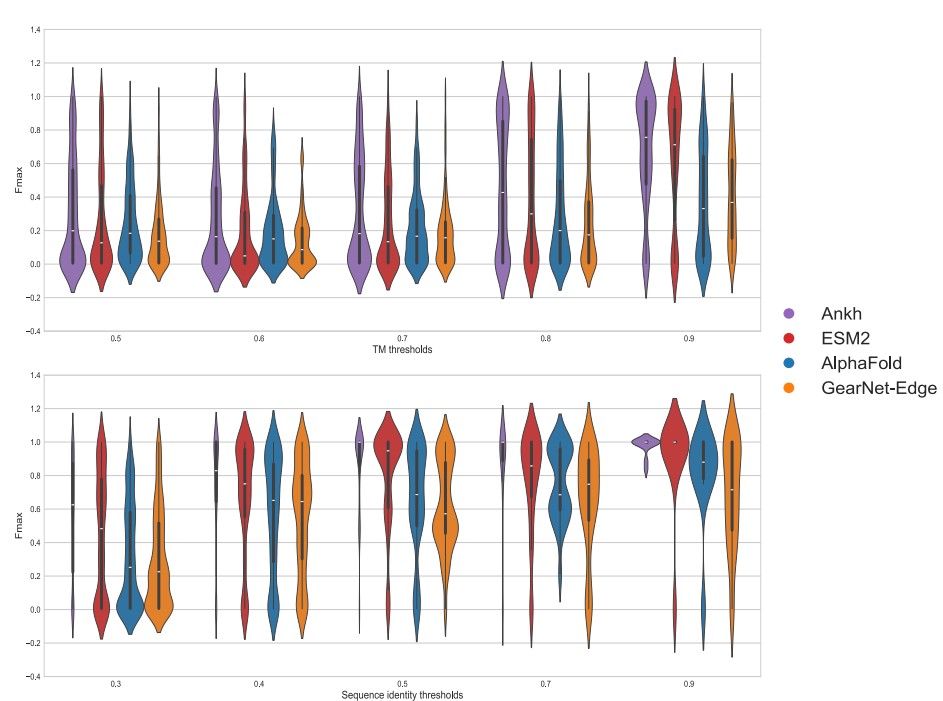

Figure 17: Distributions of $F_{max}$ and AUPCR scores for individual clusters with more than 5 members in the EC prediction task.

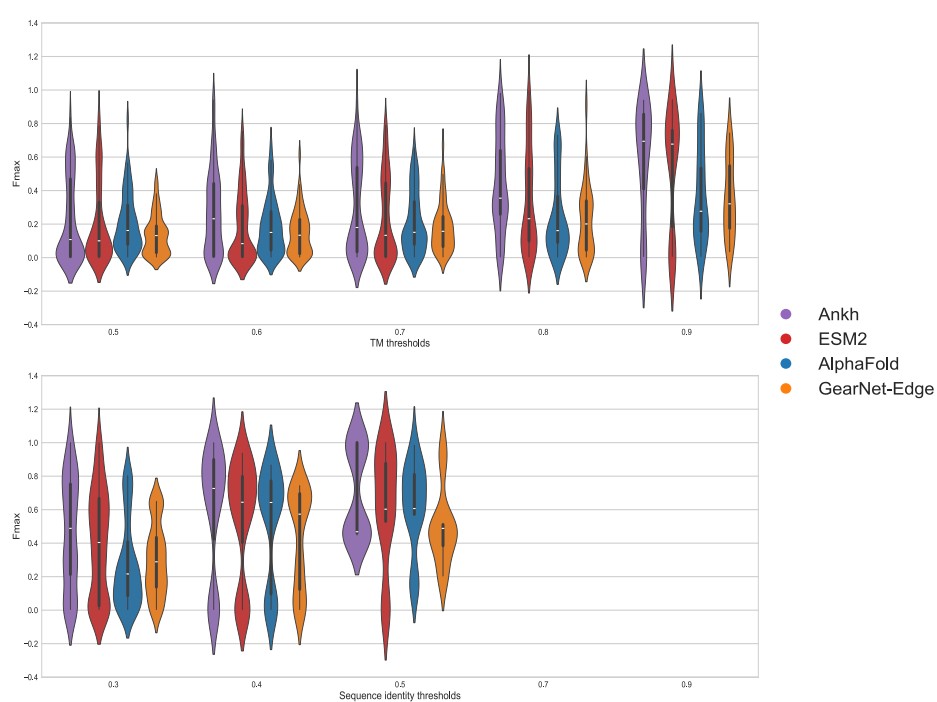

Figure 18: Distributions of $F_{max}$ and AUPCR scores for individual clusters with more than 15 members in the EC prediction task.

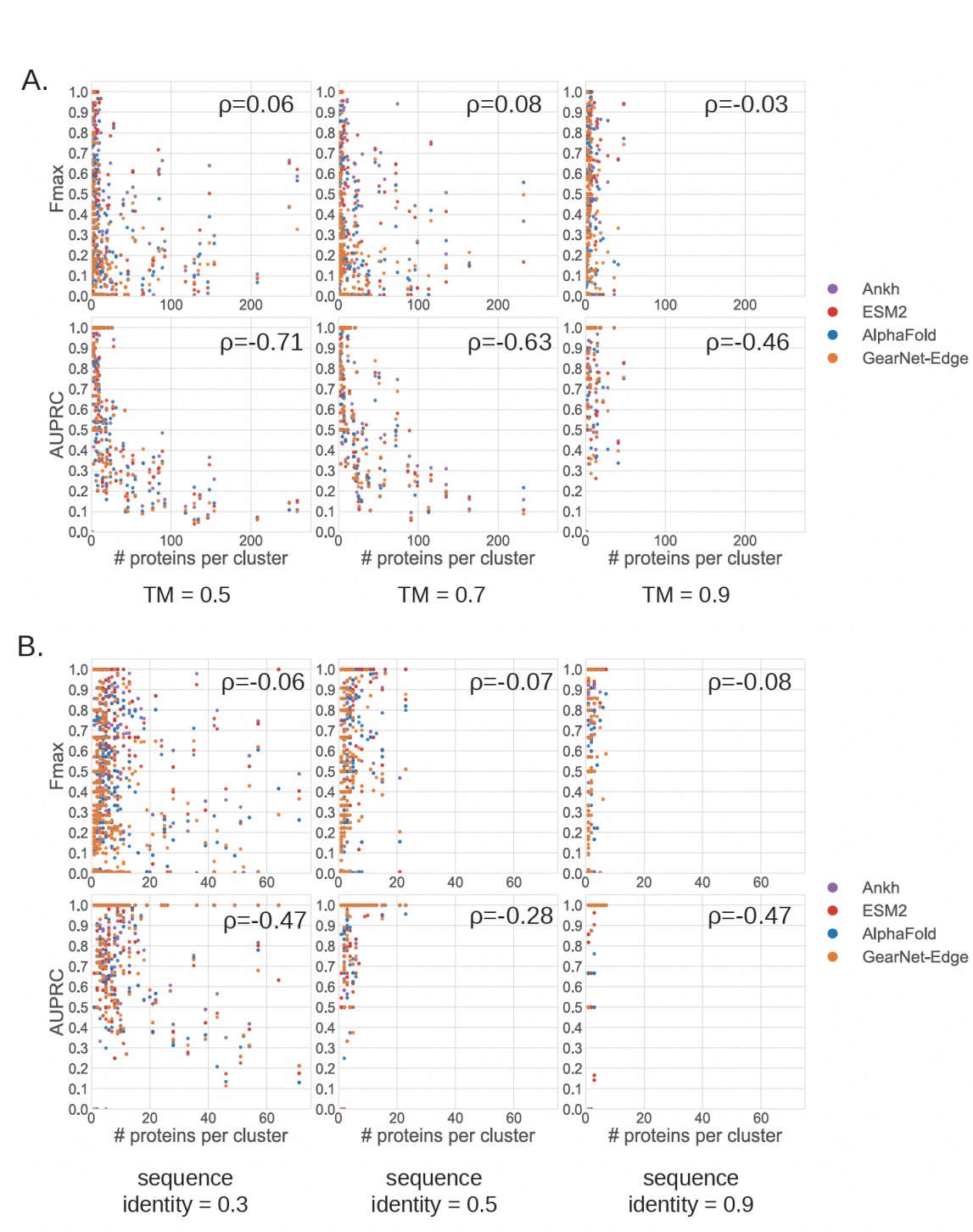

Figure 19: $F_{max}$ and AUPCR scores for individual clusters as a function of cluster size in the EC prediction task. (A): results for TM thresholds, (B): sequence identity thresholds. $\rho$ is the Spearman correlation between the score values and the cluster sizes.

# I  DETAILS ON THE DATA SPLITS

In this section, we provide a detailed description of our sequence-based and structure-based data splits.

## I.1  SIMILARITY SCORES

Pairwise sequence identities are calculated using MMseqs2 (Steinegger & Söding, 2018), which performs local alignments of sequences. TM scores between protein structure pairs are obtained using Foldseek with the TMAlign option (van Kempen et al., 2023). Foldseek first prefilters similar protein pairs based on their structural motifs and then realigns the structures with a fast implementation of TM-align for improved accuracy. We set the target coverage to 80% or both MMseqs2 and Foldseek, without specifying minimum values for sequence identity or TM score. The following parameter values are used:

- MMseqs2: `-cov-mode 1 -c 0.8 -alignment_mode 3 -evalue=0.001 -sensitivity 7.5 -min-seq-id 0`.

- Foldseek: `-cov-mode 1 -c 0.8 -alignment_mode 3 -evalue=0.001 -sensitivity 7.5 -tm_thresh 0 -lddt_threshold 0 -tmalign_fast 1 -alignment_type 1`

## I.2  DATA SPLIT STATISTICS

We provide an overview of the EC and GO data splits in Table 14, along with detailed statistics for each split in Tables 15-18. For each dataset and similarity metric, we evaluated models on 10 of the 20 sampled data splits, selecting those that best matched the 80:10:10 train:val ratio. However, we systematically include all 20 data seeds in our tables to allow users to aggregate performance over additional seeds or utilize train:val splits with varying ratios. The data seeds used in our experiments are as follows:

- **EC, sequence identity**: 1, 5, 20, 6, 2, 7, 13, 3, 12, 4
- **EC, TM score**: 3, 13, 10, 4, 8, 15, 19, 7, 17, 14
- **GO, sequence identity**: 1, 4, 2, 11, 6, 17, 3, 9, 18, 7
- **GO, TM score**: 18, 9, 1, 12, 7, 11, 5, 13, 14, 16

Table 14: Dataset splits overview

| CLASSIFICATION | SIMILARITY METRIC | THRESHOLDS | # CLUSTERS PER THRESHOLD | |
| --- | --- | --- | --- | --- |
| | | | VALIDATION | TEST |
| EC | SEQUENCE IDENTITY | 0.3, 0.4, 0.5, 0.7, 0.9 | 175 | 175 |
| | TM SCORE | 0.5, 0.6, 0.7, 0.8, 0.9 | 48 | 48 |
| GO | SEQUENCE IDENTITY | 0.3, 0.4, 0.5, 0.7, 0.9 | 350 | 350 |
| | TM SCORE | 0.5, 0.6, 0.7, 0.8, 0.9 | 100 | 100 |

Table 15: Statistics for the sequence-based EC dataset

| | | # PROTEINS | | | SPLIT RATIO (%) | | | # UNIQUE LABELS | | |
|---|---|---|---|---|---|---|---|---|---|---|
| SEED | TOTAL | TRAIN | VAL | TEST | TRAIN | VAL | TEST | TRAIN | VAL | TEST |
| 1 | 18919 | 15173 | 1799 | 1947 | 80.2 | 9.51 | 10.29 | 538 | 460 | 480 |
| 2 | 18929 | 15253 | 1811 | 1865 | 80.58 | 9.57 | 9.85 | 538 | 471 | 472 |
| 3 | 18928 | 15337 | 1777 | 1814 | 81.03 | 9.39 | 9.58 | 538 | 456 | 466 |
| 4 | 18925 | 15436 | 1823 | 1666 | 81.56 | 9.63 | 8.8 | 538 | 473 | 444 |
| 5 | 18912 | 15208 | 1798 | 1906 | 80.41 | 9.51 | 10.08 | 538 | 456 | 477 |
| 6 | 18928 | 15232 | 2038 | 1658 | 80.47 | 10.77 | 8.76 | 538 | 472 | 464 |
| 7 | 18924 | 15256 | 1832 | 1836 | 80.62 | 9.68 | 9.7 | 538 | 477 | 466 |
| 8 | 18931 | 15635 | 1551 | 1745 | 82.59 | 8.19 | 9.22 | 538 | 447 | 458 |
| 9 | 18936 | 14807 | 2211 | 1918 | 78.19 | 11.68 | 10.13 | 538 | 482 | 469 |
| 10 | 18923 | 15698 | 1567 | 1658 | 82.96 | 8.28 | 8.76 | 538 | 459 | 440 |
| 11 | 18920 | 14659 | 1985 | 2276 | 77.48 | 10.49 | 12.03 | 538 | 475 | 486 |
| 12 | 18925 | 15352 | 1666 | 1907 | 81.12 | 8.8 | 10.08 | 538 | 443 | 481 |
| 13 | 18914 | 15259 | 1933 | 1722 | 80.68 | 10.22 | 9.1 | 538 | 460 | 452 |
| 14 | 18927 | 14500 | 1900 | 2527 | 76.61 | 10.04 | 13.35 | 538 | 472 | 493 |
| 15 | 18927 | 15488 | 1732 | 1707 | 81.83 | 9.15 | 9.02 | 538 | 465 | 444 |
| 16 | 18923 | 14476 | 2262 | 2185 | 76.5 | 11.95 | 11.55 | 538 | 484 | 491 |
| 17 | 18918 | 15542 | 1647 | 1729 | 82.15 | 8.71 | 9.14 | 538 | 458 | 456 |
| 18 | 18933 | 15771 | 1465 | 1697 | 83.3 | 7.74 | 8.96 | 538 | 453 | 433 |
| 19 | 18913 | 15461 | 1606 | 1846 | 81.75 | 8.49 | 9.76 | 538 | 441 | 468 |
| 20 | 18919 | 15054 | 1981 | 1884 | 79.57 | 10.47 | 9.96 | 538 | 474 | 462 |

Table 16: Statistics for the TM-based EC dataset

| | | # PROTEINS | | | SPLIT RATIO (%) | | | # UNIQUE LABELS | | |
|---|---|---|---|---|---|---|---|---|---|---|
| SEED | TOTAL | TRAIN | VAL | TEST | TRAIN | VAL | TEST | TRAIN | VAL | TEST |
| 1 | 16839 | 11811 | 2707 | 2321 | 70.14 | 16.08 | 13.78 | 538 | 502 | 498 |
| 2 | 16771 | 14281 | 1595 | 895 | 85.15 | 9.51 | 5.34 | 538 | 461 | 367 |
| 3 | 16823 | 13245 | 1699 | 1879 | 78.73 | 10.1 | 11.17 | 538 | 479 | 467 |
| 4 | 16824 | 13794 | 1397 | 1633 | 81.99 | 8.3 | 9.71 | 538 | 428 | 456 |
| 5 | 16751 | 10659 | 2692 | 3400 | 63.63 | 16.07 | 20.3 | 538 | 504 | 509 |
| 6 | 16818 | 12555 | 2392 | 1871 | 74.65 | 14.22 | 11.12 | 538 | 494 | 474 |
| 7 | 16729 | 14135 | 1369 | 1225 | 84.49 | 8.18 | 7.32 | 538 | 436 | 409 |
| 8 | 16879 | 13896 | 1486 | 1497 | 82.33 | 8.8 | 8.87 | 538 | 418 | 441 |
| 9 | 16831 | 10996 | 3524 | 2311 | 65.33 | 20.94 | 13.73 | 538 | 521 | 489 |
| 10 | 16829 | 13148 | 1721 | 1960 | 78.13 | 10.23 | 11.65 | 538 | 455 | 475 |
| 11 | 16805 | 10776 | 2957 | 3072 | 64.12 | 17.6 | 18.28 | 538 | 510 | 503 |
| 12 | 16743 | 12477 | 2613 | 1653 | 74.52 | 15.61 | 9.87 | 538 | 502 | 481 |
| 13 | 16828 | 13736 | 1908 | 1184 | 81.63 | 11.34 | 7.04 | 538 | 467 | 415 |
| 14 | 16876 | 14338 | 1371 | 1167 | 84.96 | 8.12 | 6.92 | 538 | 422 | 418 |
| 15 | 16832 | 12938 | 1619 | 2275 | 76.87 | 9.62 | 13.52 | 538 | 452 | 491 |
| 16 | 16859 | 11075 | 3311 | 2473 | 65.69 | 19.64 | 14.67 | 538 | 519 | 502 |
| 17 | 16809 | 12619 | 1884 | 2306 | 75.07 | 11.21 | 13.72 | 538 | 474 | 493 |
| 18 | 16852 | 14471 | 1315 | 1066 | 85.87 | 7.8 | 6.33 | 538 | 420 | 393 |
| 19 | 16838 | 12725 | 1554 | 2559 | 75.57 | 9.23 | 15.2 | 538 | 454 | 501 |
| 20 | 16727 | 12461 | 2322 | 1944 | 74.5 | 13.88 | 11.62 | 538 | 494 | 480 |

Table 17: Statistics for the sequence-based GO dataset

| | # PROTEINS | | | SPLIT RATIO (%) | | | BP | | | CC | | | MF | | |
| | | | | | | | | | | # UNIQUE LABELS | | | | | |
| SEED | TOTAL | TRAIN | VAL | TRAIN | VAL | TEST | TRAIN | VAL | TEST | TRAIN | VAL | TEST | TRAIN | VAL | TEST |
|---|---|---|---|---|---|---|---|---|---|---|---|---|---|---|---|
| 1 | 36113 | 28941 | 3544 | 3628 | 80.14 | 9.81 | 10.05 | 1942 | 1900 | 1905 | 320 | 314 | 315 | 489 | 470 | 475 |
| 2 | 36128 | 29111 | 3455 | 3562 | 80.58 | 9.56 | 9.86 | 1943 | 1917 | 1918 | 320 | 319 | 315 | 489 | 481 | 478 |
| 3 | 36116 | 29281 | 3474 | 3361 | 81.07 | 9.62 | 9.31 | 1943 | 1891 | 1892 | 320 | 317 | 307 | 489 | 476 | 475 |
| 4 | 36124 | 29038 | 3463 | 3623 | 80.38 | 9.59 | 10.03 | 1943 | 1914 | 1906 | 320 | 309 | 315 | 489 | 466 | 476 |
| 5 | 36120 | 28266 | 4097 | 3757 | 78.26 | 11.34 | 10.4 | 1943 | 1917 | 1903 | 320 | 319 | 316 | 489 | 478 | 483 |
| 6 | 36097 | 28593 | 3796 | 3708 | 79.21 | 10.52 | 10.27 | 1943 | 1917 | 1911 | 320 | 316 | 314 | 488 | 467 | 483 |
| 7 | 36107 | 28442 | 3907 | 3758 | 78.77 | 10.82 | 10.41 | 1939 | 1924 | 1908 | 319 | 313 | 317 | 488 | 483 | 481 |
| 8 | 36123 | 29357 | 3241 | 3525 | 81.27 | 8.97 | 9.76 | 1943 | 1913 | 1919 | 320 | 316 | 318 | 489 | 477 | 480 |
| 9 | 36119 | 28462 | 3915 | 3742 | 78.8 | 10.84 | 10.36 | 1943 | 1924 | 1916 | 320 | 316 | 318 | 489 | 474 | 473 |
| 10 | 36136 | 29458 | 3233 | 3445 | 81.52 | 8.95 | 9.53 | 1942 | 1890 | 1906 | 320 | 317 | 316 | 489 | 480 | 478 |
| 11 | 36128 | 28676 | 3487 | 3965 | 79.37 | 9.65 | 10.97 | 1943 | 1926 | 1908 | 320 | 311 | 313 | 489 | 470 | 478 |
| 12 | 36134 | 29625 | 3260 | 3249 | 81.99 | 9.02 | 8.99 | 1943 | 1912 | 1906 | 320 | 314 | 316 | 488 | 476 | 478 |
| 13 | 36131 | 28165 | 4211 | 3755 | 77.95 | 11.65 | 10.39 | 1943 | 1931 | 1923 | 320 | 319 | 319 | 489 | 482 | 477 |
| 14 | 36119 | 27906 | 4134 | 4079 | 77.26 | 11.45 | 11.29 | 1939 | 1927 | 1911 | 320 | 319 | 317 | 488 | 478 | 479 |
| 15 | 36118 | 29454 | 3409 | 3255 | 81.55 | 9.44 | 9.01 | 1943 | 1896 | 1929 | 320 | 313 | 314 | 489 | 478 | 476 |
| 16 | 36136 | 27322 | 4734 | 4080 | 75.61 | 13.1 | 11.29 | 1940 | 1932 | 1921 | 320 | 318 | 318 | 488 | 480 | 479 |
| 17 | 36122 | 29247 | 3340 | 3535 | 80.97 | 9.25 | 9.79 | 1943 | 1912 | 1924 | 320 | 316 | 317 | 489 | 482 | 475 |
| 18 | 36119 | 29337 | 3691 | 3091 | 81.22 | 10.22 | 8.56 | 1943 | 1918 | 1917 | 320 | 318 | 313 | 489 | 479 | 479 |
| 19 | 36126 | 29466 | 3274 | 3386 | 81.56 | 9.06 | 9.37 | 1943 | 1914 | 1870 | 320 | 314 | 313 | 489 | 486 | 473 |
| 20 | 36112 | 27898 | 3780 | 4434 | 77.25 | 10.47 | 12.28 | 1942 | 1906 | 1922 | 320 | 316 | 317 | 489 | 477 | 481 |

Table 18: Statistics for the TM-based GO dataset

| | # PROTEINS | | | SPLIT RATIO (%) | | | BP | | | CC | | | MF | | |
| | | | | | | | | | | # UNIQUE LABELS | | | | | |
| SEED | TOTAL | TRAIN | VAL | TRAIN | VAL | TEST | TRAIN | VAL | TEST | TRAIN | VAL | TEST | TRAIN | VAL | TEST |
|---|---|---|---|---|---|---|---|---|---|---|---|---|---|---|---|
| 1 | 32606 | 24942 | 3289 | 4375 | 76.5 | 10.09 | 13.42 | 1942 | 1830 | 1854 | 320 | 304 | 307 | 487 | 436 | 432 |
| 2 | 32618 | 26175 | 2843 | 3600 | 80.25 | 8.72 | 11.04 | 1942 | 1836 | 1801 | 320 | 301 | 304 | 486 | 407 | 426 |
| 3 | 32507 | 27050 | 2638 | 2819 | 83.21 | 8.12 | 8.67 | 1943 | 1793 | 1770 | 320 | 291 | 301 | 488 | 414 | 407 |
| 4 | 32640 | 27242 | 2881 | 2517 | 83.46 | 8.83 | 7.71 | 1943 | 1769 | 1772 | 317 | 290 | 301 | 488 | 419 | 409 |
| 5 | 32644 | 24797 | 3588 | 4259 | 75.96 | 10.99 | 13.05 | 1942 | 1828 | 1851 | 317 | 300 | 306 | 486 | 429 | 422 |
| 6 | 32577 | 26678 | 3562 | 2337 | 81.89 | 10.93 | 7.17 | 1940 | 1868 | 1809 | 320 | 308 | 298 | 481 | 416 | 419 |
| 7 | 32476 | 27090 | 2908 | 2478 | 83.42 | 8.95 | 7.63 | 1943 | 1783 | 1756 | 320 | 297 | 300 | 489 | 403 | 398 |
| 8 | 32612 | 26479 | 2637 | 3496 | 81.19 | 8.09 | 10.72 | 1943 | 1690 | 1813 | 320 | 304 | 305 | 487 | 387 | 415 |
| 9 | 32641 | 23739 | 5374 | 3528 | 72.73 | 16.46 | 10.81 | 1937 | 1903 | 1865 | 317 | 315 | 300 | 482 | 446 | 424 |
| 10 | 32749 | 26277 | 2853 | 3619 | 80.24 | 8.71 | 11.05 | 1943 | 1810 | 1804 | 320 | 307 | 301 | 489 | 401 | 421 |
| 11 | 32578 | 22764 | 4892 | 4922 | 69.88 | 15.02 | 15.11 | 1940 | 1902 | 1891 | 319 | 302 | 308 | 481 | 434 | 448 |
| 12 | 32687 | 25556 | 2790 | 4341 | 78.18 | 8.54 | 13.28 | 1943 | 1818 | 1839 | 320 | 301 | 312 | 489 | 421 | 423 |
| 13 | 32421 | 26283 | 3535 | 2603 | 81.07 | 10.9 | 8.03 | 1942 | 1873 | 1713 | 320 | 300 | 295 | 488 | 427 | 395 |
| 14 | 32494 | 25214 | 4114 | 3166 | 77.6 | 12.66 | 9.74 | 1941 | 1858 | 1830 | 319 | 299 | 302 | 485 | 445 | 410 |
| 15 | 32439 | 25146 | 3175 | 4118 | 77.52 | 9.79 | 12.69 | 1943 | 1794 | 1809 | 320 | 290 | 296 | 489 | 390 | 428 |
| 16 | 32491 | 24510 | 4332 | 3649 | 75.44 | 13.33 | 11.23 | 1941 | 1860 | 1821 | 320 | 303 | 306 | 487 | 426 | 426 |
| 17 | 32432 | 26906 | 2746 | 2780 | 82.96 | 8.47 | 8.57 | 1943 | 1776 | 1788 | 318 | 297 | 295 | 488 | 407 | 400 |
| 18 | 32480 | 27325 | 2200 | 2955 | 84.13 | 6.77 | 9.1 | 1943 | 1777 | 1823 | 320 | 284 | 289 | 488 | 411 | 416 |
| 19 | 32540 | 25965 | 2886 | 3689 | 79.79 | 8.87 | 11.34 | 1943 | 1803 | 1821 | 320 | 306 | 306 | 486 | 404 | 421 |
| 20 | 32764 | 24676 | 3947 | 4141 | 75.31 | 12.05 | 12.64 | 1942 | 1868 | 1855 | 320 | 306 | 307 | 486 | 424 | 427 |

