# OpenReview forum: "Disconnecting The Dots: Creating Leakage-Free Protein Datasets by Removal of Densely Connected Data Points"
_ICLR.cc/2025/Conference — Submitted to ICLR 2025_

### Official Review · Reviewer_5Vmu · 2024-11-01

**Soundness:** 2
**Presentation:** 2
**Contribution:** 1
**Rating:** 3
**Confidence:** 3

**Summary:**

The paper presents a novel approach for creating data splits in biological datasets, to avoid data leakage to support generalization testing. The paper introduces a data-splitting methodology that strategically removes highly connected/central data points to produce well-balanced, disconnected clusters, which should allow for strong generalization assessment. The authors evaluate the proposed method by examining how various protein representations influence protein function prediction.

**Strengths:**

- The paper introduces a novel data-splitting methodology that strategically removes highly central data points to achieve well-separated, leakage-free clusters.
- The study rigorously tests the proposed method across different models. Although tests for other biological tasks are missing.
- Both sequence and structure based splits are considered

**Weaknesses:**

- The authors mention that the new clustering approach results in a data loss of around 12.4% under structural similarity and 1.4% under sequence similarity. This level of data exclusion is significant, I feel like a more detailed section about this tradeoff is missing from the paper.
- The paper places significant reliance on community detection for constructing data splits. However, community detection can sometimes create arbitrary groupings that may not accurately reflect biological relationships.
- The results are mainly focused on protein function prediction within specific ontologies, leaving it unclear how the method performs across other prediction tasks or with more diverse biological data.
- The authors emphasize the importance of choosing a similarity metric in the introduction but do not elaborate on the available similarity metrics (e.g., sequence identity, TM score). These metrics should be defined in the paper.
- A complexity and runtime analysis of the proposed method is missing.

Minor points:
“To address this issue, clustering of biological data prior to splitting has become standard practice in bioinformatics.” This statement would benefit from a few citations.
The reference at page 3 line 161 seems to be broken/not linked properly.
On the same line there seems to be typo for:
“... very low levels of sequence similarity (¿25% sequence identity)”
The authors probably wanted to say <25%. Similar typo also appears in line 4 for algorithms 1 & 2.

**Questions:**

- How did the authors ensure that the removed data points did not disproportionately affect the diversity or representativeness of the dataset, especially for small protein families?
- Have the authors tested the proposed method on additional biological tasks to assess its versatility and generalization across tasks?
- What is the total time complexity of the proposed method?

---

> ### Author Response · Authors · 2024-11-26
>
> We thank the reviewer for their review. We address concerns/questions line-by-line below.
>
> **Re: Level of data exclusion** \
> We appreciate the reviewer's concern about the level of data exclusion. While the reduction is significant, the final dataset size remains suitable for practical applications, and the number of points removed is much smaller than existing methods like GraphPart, particularly for low-to-intermediate difficulty levels. Additionally, our approach offers a robust way to assess model generalization, and training on the full dataset is always an option afterwards.
>
> We also agree that a discussion of the data loss trade-off could be beneficial. In preliminary experiments, we monitored the proportion of the largest cluster in the data as a function of removed top connectors. While removing only a subset of the top connectors was sufficient to obtain well-distributed clusters, we also found that this proportion declined only after most connectors had been removed. This means that, while a stopping criterion could be introduced (such as monitoring the proportion of the largest cluster in the data), the number of additional data points retained would be minimal.
>
> **Re: Reliance on community detection** \
> The reviewer is right to note that community detection can sometimes create arbitrary groupings. However, this should mostly happen with the Louvain version, which can sometimes create communities that are not well-defined or have unbalanced sizes. We use the Leiden version of the community detection algorithm, which aims to improve on the Louvain version by refining the community structure. It does so by ensuring that the communities it finds are more stable and well-defined, thus avoiding situations where communities overlap too much or are too fragmented.
>
> **Re: Additional biological tasks** \
> We agree that the manuscript would benefit from including more tasks. We are currently working on adding two structure task predictions (contact and fold prediction, from the PEER benchmark), one localization task (also from PEER), and two retrieval, homology-based task (from the DGEB benchmark). We don’t have the complete data yet but will include them in the final results.
>
> **Re: Definition of similarity metrics** \
> As we discuss in our response to reviewer TsLk , we agree that the similarity metrics should be defined in the main manuscript. We will add them in the final version. We used the following definitions:
> - sequence identity: we used MMseqs fractional identity (“fid”) score, which measures the proportion of identical amino acid matches between two aligned sequences, normalized by the alignment length.
> - TM score: a measure of global structural similarity widely used in protein structure comparison (e.g in AlphaFold). We use the definition introduced in the original paper (Zhang, Y., & Skolnick, J. (2004)) and its fast implementation by Foldseek.
>
> **Re: Complexity analysis** \
> As we mention in our responses to reviewers MFnt and QpPD, we provide an estimate of the computational complexity of our method in Section E of the Appendix. We estimate the complexity to be quadratic in the number of samples, although we suspect the actual cost may be lower due to the sparse implementation of our method. We plan to provide a more refined analysis in the final version of the manuscript.

---

### Official Review · Reviewer_QpPD · 2024-11-02

**Soundness:** 3
**Presentation:** 3
**Contribution:** 1
**Rating:** 3
**Confidence:** 4

**Summary:**

Note added on December 4: I appreciate the authors' careful replies to the reviews. However, the replies do not change my evaluation of the work.

Some of the replies highlight that the contribution to machine learning in general is limited. These include the assertion that discussing classification with a reject option is out of scope, the absence of a big-O analysis, and the fact that Algorithm 2 is due to previous research.

=============

All living things are related via evolution, so all biological objects such as proteins are related. Hence, it is impossible to create training/test data splits where test objects are unrelated to training objects. This submission proposes a new heuristic algorithm to create clusters that are reasonably unrelated to each other. Training and test sets can then be based on clusters.

**Strengths:**

This paper is written well and is understandable and sensible. Originality is fair relative to previous work, which is cited comprehensively AFAIK. Quality is good, with no obvious mistakes; so is clarity.

**Weaknesses:**

Originality, quality, and clarity are adequate. Significance is limited. First, there is no claim of significance outside computational biology, so the importance for machine learning and ICLR is weak. Second, the algorithm is heuristic and not very surprising. Third, within computational biology, the method does not lead to any actually more successful predictive method. Rather, it confirms two facts that are  well-known already: "structure-based splits present a significantly more challenging test than sequence-based splits and ... protein function prediction remains an unsolved task."

Overall, this paper should be published in a specialized computational biology venue, perhaps an ICLR workshop.

**Questions:**

054: There is an opportunity to make this research deeper and broader: formulate these three desiderata as axioms in the style of Arrow or Shapley. Then derive something like Arrow's impossibility theorem, or Shapley values.

Separately, discuss explicitly what the goals are when creating training and test sets. Is the concept of IID (independent and identically distributed) relevant or not? Is there a conflict between *measuring* generalization and *achieving* it? (There is always at least some conflict, because the best generalization is presumably achieved by training on all available data, with no data left out for testing.

Most learning methods have hyperparameters which are chosen using a validation set. Discuss three-way splits: training, validation, test.

057: For relevance to machine learning in general, cite and discuss briefly the relationship to classification with a reject option. Also discuss how the need arises outside comp bio to make training and test sets genuinely separate. Note that there is a large literature on removing leakage from benchmarks for LLMs.

083: This paper does have citations older than a few years, but would be more scholarly and thoughtful with more. For example, the temporal approach to leakage prevention was used for decades by the CASP contests, and has been used for decades in finance.

213: Algorithm 1 mentions three concepts with different names: communities C, connected components K, and clusters (line 250). Algo. 2 line 12 reveals that clusters are actually a different name for K. Still, what is the conceptual difference between communities and clusters? Why doesn't Algorithm 1 merely return C with the top connector nodes v removed?

229: Provide a big-O complexity analysis.

270: Explain the need for Algorithm 2. Why must many different thresholds be used?

Nodes and edges are removed in five different places: Algo. 1 line 12 and Algo. 2 lines 10, 14, 17, 21. Why so complicated, especially given that the algorithm is heuristic without guarantees?

448: Figure 3 shows that the biological reality is one single large connected component. Conceptually, a learning algorithm should be able to learn and exploit this reality. The picture on the right is "fake news" that may be helpful for measuring generalization, but unhelpful for *achieving* generalization. Note that domains other than comp bio also have a reality with one main connected component and many tiny components, in particular social networks such as Facebook.

Figure 18: Spearman correlations are not appropriate for data that is so obviously nonlinear.

**Details Of Ethics Concerns:**

None.

---

> ### Author Response · Authors · 2024-11-26
>
> We would like to thank the reviewer for their thoughtful comments and are happy that they found our work to be well written and of good quality. We discuss their concerns below:
>
> **Re: Scope** \
> Our focus here is more an experimental investigation. For space reasons we chose not to overly formalize our description of the method.
>
> **Re: Goals when creating training and test sets** \
> As we discuss in response to reviewer TsLk, proteins are not independent data points. They arise from evolutionary processes, often sharing high sequence similarity as a result. Random splits fail to account for this, frequently placing highly similar proteins across training, validation, and test sets. This overlap leads to performance metrics that do not reflect true generalization capabilities. This is a key challenge in biology because experimental labels are expensive and scarce, meaning researchers rely heavily on model predictions to annotate new proteins. Furthermore, well-studied proteins are the ones that are experimentally accessible – that is, only a small subset of existing proteins. On the other hand, proteins evolutionarily distant from the known protein universe are discovered every day.
>
> As a result, annotated datasets typically only represent a fraction of the functional protein space. This motivates the need for benchmarking methods that enable generalization assessment to distant protein families. Two notable examples of label scarcity include:
> - the human microbiome – many gut bacteria cannot be cultivated, and therefore cannot be labeled functionally;
> - protein structures – historically, structure have been determined using X-ray crystallography, which works well for soluble proteins but not for membrane proteins. For a long time, the set of structures available for training protein structure prediction models was thus limited to a small subset of protein families, making it very hard for model to generalize to new structures. Significant improvements on the modeling side were made possible by the construction of structurally diverse datasets by CASP.
>
> **Re: Measuring vs achieving generalization** \
> Yes, there is this tension in general. But even in instances where the desire is to achieve full generalization by training on all available data, it is still useful to understand the behavior of the model by using smaller subsets of the data to assess its generalization capabilities. Furthermore, in the biological context, data is often scarce. It is infer trend lines of how a give model improves as a function of the dataset size. Smaller subsets can be useful to infer how a model would improve as a function of the dataset size.
>
> **Re: Three-way splits** \
> Our approach already uses a three-way splitting, with a validation set for hyperparameter tuning and a test set for model testing. Both the validation and test sets were constructed using our leakage-free clustering approach.
>
> **Re: Classification with a reject option** \
> We appreciate the pedagogical value of this type of discussion but think they are out of the scope of this paper.
>
> **Re: Citations** \
> We have now included references to CASP in the manuscript.

---

> ### Author Response · Authors · 2024-11-26
>
> **Re: Clarifying concepts**\
> We acknowledge that the different terms used in our manuscript may be confusing, as they all refer to various notions of clustering. To clarify, we will standardize the terminology in the camera-ready version and provide clear definitions:
> - **Clusters**: Groups of nodes in a graph with high internal connectivity relative to the rest of the network, typically defined by fixed criteria for local density or similarity. Cluster-based methods, such as k-means or hierarchical clustering, are used when the groups are expected to be well-separated and internally compact.
> - **Connected Component**: A maximal subset of nodes where each node is reachable from any other node within the subset. Connected components are clusters that are strictly separated.
> - **Community**: a subset of nodes more densely connected to each other than to nodes outside the subset, emphasizing modularity. Community-based methods are typically used when analyzing large-scale networks, such as social networks or protein interaction networks. They work well with graphs, focusing on connectivity patterns. Detecting communities allows us to identify groups of highly connected proteins within the large cluster found by hierarchical clustering or connected components.
>
> In our manuscript, the resulting connected components from Algorithm 1 are the clusters we use to construct the new splits. We will revise the manuscript to use the term "cluster" consistently going forward.
>
> Additionally, Algorithm 1 does not simply return the communities C with the top connector nodes v removed. We have observed that returning the connected components leads to slightly more clusters compared to returning communities. This occurs when the removal of v splits a community into two separate components. While this does not affect the total number of nodes removed, it results in a greater number of clusters to sample from when creating the splits, thereby enhancing the diversity of the clusters.
>
> **Re: Complexity analysis** \
> As we mention in our response to reviewer MFnt, we provide an estimate of the computational complexity of our method in Section E of the Appendix. We estimate the complexity to be quadratic in the number of samples, although we suspect the actual cost may be lower due to the sparse implementation of our method. We plan to provide a more refined analysis in the final version of the manuscript.
>
> **Re: Algorithm 2** \
> We use Algorithm 2 to evaluate model performance at different difficulty levels. These often correspond to distinct biological problems, such as de novo design (high difficulty, requiring generalization) and protein optimization (low difficulty, requiring less generalization).
> However, we recognize that Algorithm 2 adds complexity to the main manuscript. It implements an existing method for distributing clusters across training, validation, and test sets to create multiple difficulty levels. This method, along with the ProteinNet dataset, was introduced five years ago and has been used in several protein benchmarks. We present a sparse, fast implementation to apply it to new datasets, including our DeepFRI splits. The core contribution of our work, however, is the removal of leakage through community detection, as described in Algorithm 1. To clarify this, we will move the section and figure discussing Algorithm 2 to the Appendix in the final version of the manuscript.
>
> **Re: Spearman correlation** \
> We agree that Spearman correlations are not appropriate for that type of data, and will remove it in the final version.

---

### Official Review · Reviewer_TsLk · 2024-11-02

**Soundness:** 2
**Presentation:** 2
**Contribution:** 2
**Rating:** 3
**Confidence:** 4

**Summary:**

The paper addresses the issue of random splitting data in the context of cross validation for predictive tasks. The paper described the problem as data leakage and inflating reported generalization performance. The paper suggested the strategy as clustering data and then splitting among clusters. The paper described the root cause of the issue being biological data are generated by a universal evolutionary process that connects all known life on Earth and aims to improve fidelity of model assessment.

**Strengths:**

The motivation was very attractive, to address potentially inflated reported model generalization. And the paper attempted to solve the problem in generic way, ie. not specific to a given predictive task, or not specific to a type of data.
The proposed technique of detecting communities by removing highest degree of nodes is neat and well-motivated.

**Weaknesses:**

I found the motivation lacking evidence, how reported model generalization has been inflated. The paper tried to address a very challenging problem, ie. improve fidelity evaluation methods for generic predictive tasks. I expect the paper to provide more rigorous design how to quantify that the proposed scheme can make evaluation more trustworthy, or model performance generalizes better. Some notion of defining and quantifying model generalization performance would be useful to support the value of the proposed method.

In addition, I think the paper would be stronger if the paper can clarify, why being generated from evolutionary processes can be a problem when we split the data randomly – one suggestion would be to motivate from the specific tasks, for example, when a researcher needs high fidelity prediction of protein functions and has various tools in hand – how reported model performance can help the researcher.

**Questions:**

Line 085: it is not clear why the temporal split is a problem for actual applications.

Line 161: missing reference and format errors

Line 193: some clarification would be useful to justify the assumption of an appropriate similarity metric exists; especially, such information used to calculate such similarity metric needs to not introduce data leakage as well for predictive tasks.

Line 235: format error

Line 366, Line 370, Line 519, I didn’t quite follow what results/metrics are relevant to support the claim that the proposed dataset construction pipeline provides an ideal framework for protein representation.

---

> ### Author Response · Authors · 2024-11-26
>
> We thank the reviewer for their review. We are glad they believe our approach of detecting communities “neat and well-motivated”. We have updated the manuscript to fix the formatting errors. Below, we address the reviewer’s questions and concerns individually.
>
> **Re: Lack of evidence & relevance** \
> In response to the reviewer’s comment about a lack of direct evidence, we acknowledge that the results section only indirectly demonstrated the benefit of our approach. To address this, we have made the following changes to more clearly show the advantages of our method:
> - In Section A of the Appendix, we quantify leakage in the DeepFRI splits.
> - In Section C of the Appendix, we compare model performance on the DeepFRI and our splits. We find that results are more sensitive to similarity thresholds with our splits, which is desirable for assessing model generalizability.
> We plan to incorporate both sections into the main body of the manuscript in the final version.
>
> **Re: Random and temporal splits** \
> Proteins are not independent data points. They arise from evolutionary processes, often sharing high sequence similarity as a result. Random splits fail to account for this, frequently placing highly similar proteins across training, validation, and test sets. This overlap leads to inflated performance in expectation, as models can rely on memorizing patterns rather than generalizing to novel proteins. For tasks like protein function prediction, where researchers need reliable predictions for newly discovered, evolutionarily distant proteins, this is a significant limitation. Reported performance from random splits does not reflect the real-world generalization ability of models, making it harder for researchers to choose the most effective tools for annotating novel proteins.
>
> Temporal splits face a similar issue: they also do not account for protein similarity, and can result in biologically close proteins being partitioned across dataset splits. While temporal splits might seem meaningful for applications where time order is relevant, they fail to address the key challenge in biology: generalizing to novel proteins that are highly dissimilar from known proteins. This is especially critical because experimental labels are expensive and scarce, meaning researchers rely heavily on model predictions to annotate new proteins. Evaluating models with similarity-based splits ensures that reported metrics realistically reflect their utility in these challenging scenarios.
>
> **Re: Similarity metric** \
> As we discussed in our response to reviewer MFnt, we report results for two standard similarity metrics for proteins:
> - sequence identity: we used MMseqs fractional identity (“fid”) score, which measures the proportion of identical amino acid matches between two aligned sequences, normalized by the alignment length.
> - TM score: a measure of global structural similarity widely used in protein structure comparison (e.g in AlphaFold). We use the definition introduced in the original paper (Zhang, Y., & Skolnick, J. (2004)) and its fast implementation by Foldseek.

---

### Official Review · Reviewer_MFnt · 2024-11-03

**Soundness:** 3
**Presentation:** 3
**Contribution:** 2
**Rating:** 5
**Confidence:** 4

**Summary:**

The paper is about splitting the data into training, validation, and test sets when a data set is not IID. The paper focuses on protein sequence data sets, where all existing sequences are sampled through evolution and related to each other. The paper acknowledges that this problem has been known in bioinformatics and that a typical way to address it is by creating validation/test sets such that their examples are not similar (using specific similarity thresholds) to examples in training data. The claim is that existing data splitting approaches in the protein domain do not find a good tradeoff between ability to evaluate model generalization and having enough representative data left for training. To address it, the paper proposes a new splitting method that identifies communities of proteins and removes proteins that are connectors between those communities. The new approach is evaluated on several representative protein prediction tasks.

**Strengths:**

+ The main strength of the paper is in a new way to split protein data sets aided by identifying communities of proteins. One the communities are identified and central proteins linking those communities are removed from data, it is possible to generate cleaner training-test splits that results in more diverse sets of training proteins. The proposed approach is well justified and sound.
+ The experimental evaluation is quite extensive, providing insights into the benefits of the proposed approach compared to the state of the art

**Weaknesses:**

- The main weakness of this paper is that this is a relatively niche area for the ICLR community. While the contribution might be of practical interest to the bioinformatics research community, the methodological contribution (finding communities and removing central proteins prior to data split) is relatively minor for the machine learning community.
- A weakness related to the community finding approach is that it does not scale well (what is the computational cost, is it cubic in the number of examples?). As a results, the paper only evaluates the proposed splitting approach on relatively small data set (up to 30,000 proteins), although protein sequence data sets often go into millions.
- The paper does not provide convincing experimental results that the proposed split makes a large practical difference. The arguments provided are mostly of the qualitative nature.
- The paper claimed that the approach is "agnostic to the splitting strategy is the choice of similarity metric". This is an issue because the similarity metric is a key choice that can greatly impact splitting model training and evaluation. It would be good to see the evaluation using at least two different types of similarity metrics.
- The paper aims to improve on deepFRI partitioning, which does not remove central proteins after clustering (finding communities). But, deepFRI is only one of the existing splitting approaches, and one of the rare ones that results in leakage between training and test data. There are many other splitting methods that do not result in leakage.
- The concept of leakage is never clearly defined in the paper. It would be very useful to see what it means mathematically early in the paper.

**Questions:**

Please see the weaknesses.

---

> ### Author Response · Authors · 2024-11-26
>
> We thank the reviewer for their constructive feedback. We are pleased that the reviewer finds our data splitting approach to be “justified and sound” and our experimental evaluation extensive. We attempt to address remaining concerns/questions line-by-line below.
>
> **Re: Venue Fit**\
> While we acknowledge that our manuscript is most valuable to practitioners of machine learning in bioinformatics, we believe the most recent revision of our manuscript is a good fit for ICLR for three reasons:
> 1. The bioinformatic machine learning community is large and growing, especially in the wake of AlphaFold2. For instance, at ICLR 2024, there were 19 papers with “protein” in the title, 23 papers with “molecule” or “molecular”, and another 2 with “biological”. One of these introduces a benchmark for DNA language models on 6 biological tasks (“BEND: Benchmarking DNA Language Models on biologically meaningful tasks”). Previous protein-related ICLR papers have been very successful with hundreds of citations (e.g. "Transformer protein language models are unsupervised structure learners" in 2021, “Independent SE(3)-Equivariant Models for End-to-End Rigid Protein Docking" in 2022, or “Diffusion probabilistic modeling of protein backbones in 3D for the motif-scaffolding problem" in 2023 ). Furthermore, the official ICLR call for papers includes “applications in biology”. In general, as ML/AI in science has grown, biology and in particular molecular biology have been one of the key application areas.
> 2. The problem of similarity between the training, validation, and test set is crucial to estimating model generalizability in biology. The approach and the datasets we introduce should be of interest to many machine learning practitioners in biology, precisely because biological problems introduce new challenges that are not present to the same degree in vision or natural language
> 3. Although we have not explored applications in other fields of machine learning, the approach could find interesting applications elsewhere, particularly in areas where there is a well-justified similarity metric, for example in chemistry and materials science.
>
> **Re: Scaling**\
> With regards to whether our community-based approach scales well to large datasets, we provide an estimate of the computational complexity of our method in Section E of the Appendix. We estimate the complexity to be quadratic in the number of samples, although we suspect the actual cost may be lower due to the sparse implementation of our method. We plan to provide a more refined analysis in the final version of the manuscript. Additionally, our choice of dataset was driven by the task at hand and the availability of labels, rather than dataset size. The exemplar dataset we selected happened to be relatively small, which is a common scenario in biology. However, due to the sparse implementation of our method, it should also be applicable to larger datasets.
>
> **Re: Qualitative results**\
> We recognize that the original manuscript focused primarily on the comparative performance of models on the new sequence- and structure-based splits. In Section C of the Appendix, we now compare model performance on the DeepFRI and our splits. We observe that results are more sensitive to similarity thresholds with our splits, which is desirable for assessing model generalizability. We plan to move this section to the main body of the manuscript in the final version.
>
> **Re: Choice of similarity metric**\
> You are right that our rewording might have been misleading. We meant that the choice of different similarity metrics is independent of our splitting strategy, not that they would have no effect on the outcome - they certainly would. Furthermore, we already report results for two commonly used protein similarity metrics: the sequence identity (for sequences), and the TM score (for structures). Figure 4 in the main paper presents results on the new splits using both similarity metrics. Additionally, Section B of the Appendix displays the cluster distribution for both metrics, both before and after applying our method.

---

> > ### Comment · Reviewer_MFnt · 2024-12-02
> > **Reaction to the author response**
> >
> > I acknowledge that I read the response to my review and I also looked at other reviews and their responses. I thank the authors for their thoughtful comments and for adding more information to the paper. However, I still think that splitting of protein sequence data  into train and test subsets  is a relatively niche research problem that is more suitable for bioinformatics venues. The cited bioinformatics-related papers recently published in ICLR are mostly about new deep learning approaches, which is more closely related to the core interests of the conference. As a result, I prefer not to change the original ratings for this paper.

---

> ### Author Response · Authors · 2024-11-26
>
> **Re: Splitting methods and leakage** \
> The common approach for creating similarity-based splits involves clustering proteins at a set similarity threshold, then distributing these clusters into training, validation, and test sets. CD-HIT and MMseqs are widely used for this purpose. For example, CD-HIT was used to create the DeepFRI splits, while MMseqs was employed to generate the TAPE, ProteinFlow, and ProteinShake benchmarks. MMseqs also helped create the dataset for the inverse folding model ProteinMPNN. However, neither method ensures strict cluster separation by default, allowing for pairs of proteins with similarity higher than the specified threshold across clusters. As shown in Section A of the Appendix, this leads to significant leakage in the DeepFRI splits. While MMseqs offers a “connected components” mode to ensure strict separation, it produces a single large cluster, making it difficult to create balanced and diverse splits.
>
> **Re: Concept of leakage** \
> We agree that a definition of leakage is necessary. Following previous work, we define leakage as the presence of proteins in the validation (or test) set that are more similar to the training set than they should be, given the specified similarity threshold. In Section A of the Appendix, where we discuss leakage in the DeepFRI splits, we quantify it by calculating the proportion of such proteins in the validation (or test) set. We plan to move this section to the main body of the manuscript in its final version.

---

### Meta-Review · Area_Chair_BY9G · 2024-12-15

**Metareview:**

The paper is primarily concerned with the problem that protein structures or sequences should not be treated as IID samples. It investigates threshold mechanisms for identifying data leakage and subsequently uses that for creating Test and Training Datasets for Machine Learning on Proteins(both sequence and structure). The concern is limited methodological contributions to the field of Machine Learning. There is consensus that ICLR mayn't the best venue for this kind of work.

**Additional Comments On Reviewer Discussion:**

The authors, during the rebuttal, tried to address the concerns. By and large the referees were not convinced and there was consensus that the paper maybe recommended for reject.

---

### Decision · Program_Chairs · 2025-01-22

Reject